# Individual differences in working memory impact the trajectory of non-native speech category learning

Casey L. Roark[1,2]\*, Giorgio Paulon[3], Giovanni Rebaudo[3], Jacie R. McHaney[1], Abhra Sarkar[3], Bharath Chandrasekaran[1,2]\*

**1** Communication Science & Disorders, University of Pittsburgh, Pittsburgh, PA, United States of America, **2** Center for the Neural Basis of Cognition, Pittsburgh, PA, United States of America, **3** Statistics and Data Sciences, University of Texas at Austin, Austin, TX, United States of America

\* casey.roark@unh.edu (CLR); Bharath.chandrasekaran@northwestern.edu (BC)

**Data Availability Statement:** The stimulus materials, data, and analysis code are publicly available through the Open Science Framework

## Abstract

What is the role of working memory over the course of non-native speech category learning? Prior work has predominantly focused on how working memory might influence learning assessed at a single timepoint. Here, we substantially extend this prior work by examining the role of working memory on speech learning performance over time (i.e., over several months) and leverage a multifaceted approach that provides key insights into how working memory influences learning accuracy, maintenance of knowledge over time, generalization ability, and decision processes. We found that the role of working memory in non-native speech learning depends on the timepoint of learning and whether individuals learned the categories at all. Among learners, across all stages of learning, working memory was associated with higher accuracy as well as faster and slightly more cautious decision making. Further, while learners and non-learners did not have substantially different working memory performance, learners had faster evidence accumulation and more cautious decision thresholds throughout all sessions. Working memory may enhance learning by facilitating rapid category acquisition in initial stages and enabling faster and slightly more careful decision-making strategies that may reduce the overall effort needed to learn. Our results have important implications for developing interventions to improve learning in naturalistic language contexts.

## Introduction

Categorization involves mapping variable inputs to discrete labels and is an important process that supports complex cognitive processes, such as object recognition [1] and speech perception [2]. Humans can learn novel categories throughout the lifespan across different perceptual modalities. However, there are also large individual differences in the underlying learning processes and outcomes [3, 4]. As such, there is a need to better understand what contributes to successful or less successful learning. In this study, we systematically examine the

repository and can be accessed online at https://doi.org/10.17605/OSF.IO/WDPYU.

**Funding:** This research was supported by the National Institute on Deafness and Other Communication Disorders [R01DC013315A1 to B. C., F32DC018979 to C.L.R., and T32DC011499 to K. Kandler and B. Yates (trainee: J.R.M.)] and the National Science Foundation [NSF-1953712 to B.C. & A.S.]. The funders had no role in study design, data collection and analysis, decision to publish, or preparation of the manuscript.

**Competing interests:** The authors have declared that no competing interests exist.

contributions of an ability that has been linked to category learning in prior work–working memory capacity.

Working memory (WM) reflects the resources available for the temporary storage and manipulation of information relevant for a given task [5, 6]. Category learning involves many processes that are dependent on WM. Learners need to attend to task-relevant features and ignore task-irrelevant features, maintain features of a stimulus in mind as relevant or irrelevant for a decision, hold hypotheses in mind about stimulus-category-response mapping, compare representations of the stimulus to previous stimuli or rules, and incorporate feedback to update existing category representations and hypotheses about category identity. The ability to learn categories across sensory modalities has generally been found to be positively associated with WM [7–12]. WM is thought to support faster initial category learning [7] by allowing learners to hold multiple hypotheses about category identity at mind and test these hypotheses and specifically to rapidly and efficiently find a useful hypothesis [13].

Importantly, prior studies have primarily focused on the role of WM in initial learning, and, as a result, it is unclear how WM may play a role in maintenance of performance or learning patterns over time. In the earliest stages of learning, learners must be highly flexible with their behavior and search a large pool of potential hypotheses about category identity. As performance improves and becomes more stable over time, WM processes may be less relevant because learners may be making small refinements to existing rules rather than keeping many competing hypotheses in mind. As a result, it is necessary to examine learning beyond very initial learning especially for categories that are difficult or challenging to acquire within a single session.

In the current study, we examine a specific case of category learning that is an important skill in second language acquisition–non-native speech category learning. The ability to learn a new language has been positively associated with individual abilities like WM capacity [14–17]. Assessed in a single session, the ability to learn sounds of a non-native language in adulthood has been positively linked to WM capacity [11, 12]. However, other studies examining learning across longer training periods (e.g., multiple sessions across many days) have found that WM ability does not predict the ability to learn non-native speech categories [18, 19]. The role of WM across the trajectory of non-native speech category learning is not yet clear. It is possible that WM supports initial, but not later speech learning.

In the current study, we train participants on non-native Mandarin tone categories. In Mandarin, distinct pitch patterns are lexically contrastive–the same syllable produced with four different pitch patterns (e.g., high-flat, low-rising, low-dipping, and high-falling) alters the meaning. Learning to distinguish sounds based on these pitch patterns can be difficult for non-native listeners and there are large individual differences in learning [3, 20–24].

For both speech and artificial perceptual categories, training beyond one session can be very successful, leading to significant learning and retention over time. In studies not focused on WM, participants learn through extensive training over several weeks [24–29] and then sometimes are brought back for a test of retention months later (e.g., [27]– 3 months; [24]– 8 weeks). Neural representations of categories rapidly emerge within a single session of initial learning [30, 31], but continue developing over time with more experience [24].

The role of WM beyond initial category acquisition is not well understood. Whereas initial learning involves testing a large range of possible hypotheses about stimulus-response mapping and using feedback to update these hypotheses, learning beyond the novice stage involves refining existing hypotheses, learning about idiosyncratic stimuli, and continuing to develop and refine representations. Additionally, after a delay in experience, learners must reactivate existing representations and hypotheses to continue refining their category knowledge. It is possible that these processes rely less on WM than initial testing among multiple hypotheses as

is necessary during initial learning. In the current study, we examine the role of WM in both an initial learning session and learning sessions after one and three months from the initial session.

Our approach involves inviting participants who previously completed a single session of training on Mandarin tone categories [12] back for additional training sessions. McHaney et al. [12] demonstrated that WM abilities were related to success in initial non-native speech category learning across two experiments–one behavioral (revisited here) and one with pupil-lometry. Specifically, individuals with higher WM capacity were better at learning, better at finding task-appropriate strategies, and had pupil responses that reflected better stimulus-related attention. Based on this, McHaney et al. [12] concluded that WM may support learning by enhancing attention to task-relevant information. Critically, because this prior study tested only a single session of learning, it is possible this conclusion may only apply to initial learning. In the current study, we invite participants from Experiment 1 of McHaney et al. [12] back for two additional sessions–one session one month after their initial training and another session two months after the second session. We follow up with the same sample from McHaney et al. [12] to understand how individual differences in WM relate to individual differences in learn-ing beyond initial acquisition.

An important aspect of understanding individual differences in learning is the acknowl-edgement that many individuals perform at chance levels even after extensive training. We see two main possibilities that could explain this pattern–(1) these participants are actively engaged and trying but persistently fail to learn and/or (2) these participants are actively disen-gaged and are not trying to learn, so they fail to learn. Disentangling these two possibilities is challenging. Prior work on category learning takes one of two approaches regarding partici-pants performing at chance levels. Some studies remove these participants entirely, typically by removing participants who perform at or below chance levels by the end of learning [32–35]. Other studies retain these participants in the sample as it is impossible to know if their performance reflects a true inability to learn or whether they are disengaged [36–39]. The lack of consistency in these approaches across studies makes it difficult to understand this poor per-forming subset of the population. In the current study, we take a hybrid version of these approaches to better understand the underlying challenges facing less successful performers. We examine both the entire set of participants and participants who perform at above-chance levels (i.e., learners vs. non-learners who do not perform at above-chance levels). By examining the patterns while considering if participants eventually learned or not, we can better under-stand behaviors and abilities that lead to success.

We employ a multifaceted approach to understand what WM does or does not do for initial and later learning of speech categories. Specifically, we assess if WM is related to (1) perfor-mance in initial and later learning sessions, (2) maintenance of category knowledge over time, (3) generalization of category knowledge to different talkers, (4) rate of evidence accumulation and (5) response caution during decision making (Table 1).

## Initial and later learning

Based on prior work, we expect that higher WM will be beneficial to initial acquisition (i.e., session 1) of non-native speech categories [12]. This prediction stems from prior work that has demonstrated that higher WM is associated with faster and better initial artificial category learning [7–9, 11, 40–42]. We also expect to observe this pattern given that the first session of training was published in McHaney et al. [12] where among all 195 participants, WM was pos-itively related to learning. A subset of these participants (107/195) returned for the current study.

**Table 1. Hypothesized role of working memory across measures.**

| | Hypothesized role of working memory | Relevant measure(s) in current study |
|---|---|---|
| Initial learning | Hold multiple hypotheses in mind, better and faster learning | Accuracy in session 1 |
| Later learning | Enhanced attention and motivation | Accuracy in sessions 2 and 3 |
| Maintenance of category knowledge | Quickly reactivate and flexibly use existing representations | Accuracy in first block of sessions 2 and 3 compared to final block of sessions 1 and 2 |
| Generalization | Flexibly apply rules to new contexts | Accuracy in generalization test with different talkers and no feedback |
| Evidence accumulation | More efficient processing, mobilization of attentional resources | Evidence accumulation (drift) rate parameter from drift diffusion modeling |
| Response caution | More cautious, gather more information and test against multiple hypotheses before a decision is made | Decision threshold (boundary) parameter from drift diffusion modeling |

Working memory capacity is operationalized by the operation span score (OSPAN).

Building on the prior study, we will probe the extent to which WM is related to performance in subsequent learning sessions. It is possible that WM provides benefits only in initial learning by quickly allowing learners to test many different hypotheses and find the ones that maximize their performance (e.g., [13]) and that WM is unrelated to learning beyond this novice stage. This prediction would be consistent with the observation that WM is not related to speech category learning when assessed after eight days of training [18, 19]. However, it is also possible that WM provides benefits to learning beyond initial acquisition, allowing for enhanced further refinement of category representations.

## Maintenance of category knowledge

By probing performance after one and two months of no additional exposure or training, we will examine the maintenance of performance over time. One possibility is that higher WM may allow learners to quickly reactivate and flexibly use their category representations developed in prior session(s). However, it is possible that maintenance of performance over time may be independent of WM and could reflect long-term memory abilities instead.

## Generalization

The ability to accurately identify novel category exemplars is a hallmark of categorization. We will assess generalization in each session by presenting learners with novel stimuli spoken by novel talkers that they do not encounter during training, without providing feedback about the correct category. To successfully generalize to these novel talkers, they will need to apply their existing knowledge flexibly to the new context. It is possible that generalization relies on WM, as the ability to flexibly apply rules (e.g., cognitive flexibility) is correlated with WM capacity [43, 44] and generalization to novel contexts is related to individual differences in WM capacity [45, 46].

## Decision processes during learning

Using a drift diffusion modeling (DDM) approach [47, 48], we will examine whether different components of the decision process (e.g., rate of evidence accumulation and response caution) are related to WM. DDMs are popular tools to understand decision making processes from accuracy and response time measures [49–53]. DDMs assume that during decision making, sensory evidence for multiple decision alternatives is accumulated in the human brain at varying rates, and a decision is made when such evidence reaches a particular boundary [47, 48].

In the case of learning non-native speech sound categories like Mandarin tone categories, as a participant hears a stimulus, they begin accumulating evidence towards all four response options (e.g., high-flat, low-rising, low-dipping, high-falling). Each of the four response options has its own decision threshold, with higher thresholds requiring more evidence to be accumulated before the decision will be made, reflecting more cautious responding. Evidence is also accumulated toward each threshold at its own rate, with faster rates reflecting higher quality of evidence extracted from the stimulus. Below we consider the possibility that WM relates to these two components of the decision process.

The classical literature on DDMs has focused almost exclusively on binary decision-making in static settings and typically focuses on group-level analyses rather than heterogeneity across individuals. Recently, Paulon et al. [53] extended these models significantly, accounting for situations with more than two decision alternatives, heterogeneity across individuals, and longitudinal evolution of the decision-making processes by considering individual-specific and time-varying accumulators of evidence. As such, we will examine decision processes over time with estimates at both the group and *individual subject level*.

**Rate of evidence accumulation.**   We predict that more WM resources may enable learners to acquire information from the stimulus more quickly, thereby reducing the perceived difficulty of the task and effort needed to learn. The rate of evidence accumulation reflects the quality of information extracted from the stimulus, with faster rates reflecting a faster evidence accumulation process. The evidence accumulation process may also reflect efficiency of retrieval or access to categorization exemplars or other representations in memory. Faster evidence accumulation rates are associated with motivation and better task performance [54]. Prior work has demonstrated that evidence accumulation rates are related to WM abilities, with faster evidence accumulation associated with higher WM capacity [55, 56].

**Response caution.**   We predict that more WM resources may allow learners to be more cautious and less impulsive in their responses and to collect more evidence for a particular category response before making a decision. Response caution is reflected in the decision threshold. Higher thresholds reflect more cautious responses that need more evidence before a decision is made, whereas lower thresholds reflect more impulsive responses based on less evidence [57]. More difficult tasks result in more cautious response patterns, requiring that participants gather more information to make decisions [48, 58]. Individuals with higher WM capacity may have sufficient resources to gather and consult more information during decision making. As such, they may be more cautious in their responses, gathering more information to hold in WM as they learn to make more accurate decisions. This may ensure that the learner builds up enough of a representation of the stimulus before they make a response and, thus, enhance learning. Alternatively, individuals with higher WM capacity may have sufficient resources to maintain similar decision thresholds as individuals with lower WM capacity, enabling them to respond faster without making sacrifices in accuracy.

## Summary

To summarize, we examine the relationship between WM capacity and non-native Mandarin tone speech category learning in an extended training task with three sessions separated by one and two months, respectively. To gain mechanistic insights on the putative relationship between WM and individual differences in category learning over time, we assess behavior from multiple angles. Specifically, we examine how initial and later learning performance, maintenance of performance across delays, generalization to novel talkers, rate of evidence accumulation, and response caution are related to WM capacity (Table 1).

## Methods

Participants completed three sessions of Mandarin tone category learning separated by at least one and two months (Session 1 to 2: $M$ = 32.1 days, $SD$ = 0.68, range 31.7–35.6 days; Session 2 to 3: $M$ = 61.4 days, $SD$ = 2.56, range 56.6–70.9 days). Data from the first session appeared in a previously published study [12], and the second and third sessions have not appeared elsewhere.

### Participants

Participants were adults ages 18–35 recruited from Prolific (prolific.co) and participated via Gorilla Experiment Builder [59]. A total of 198 participants completed session 1 (99 Female (F), 99 Male (M), $M$ = 25.0 years, $SD$ = 4.97). Three participants were excluded because they did not follow instructions on the WM task, leaving a total of 195 participants in session 1 (98 F, 97 M, $M$ = 24.9 years, $SD$ = 4.89). There was substantial attrition from session-to-session, and we excluded participants who did not complete all sessions– 153 completed session 2 (70 F, 83 M, $M$ = 24.9 years, $SD$ = 5.05), and 107 completed session 3 (47 F, 60 M, $M$ = 24.8 years, $SD$ = 5.07). Participants who completed only one or two sessions did not differ in WM or categorization accuracy compared to those who completed all sessions (Fig A in S1 File).

Participants completed a language history questionnaire prior to participating. All participants were native speakers of non-tonal languages and reported no prior experience with any tonal languages, including Mandarin. Participants were given a sound check before the start of each session to ensure they could hear the sounds and were wearing headphones. Participants received $10/session for their participation (total up to $30 across three sessions). Informed consent was obtained from all participants. The study protocol was approved by the Institutional Review Board at the University of Pittsburgh.

### Stimuli

The stimuli were natural speech productions recorded from four native speakers (2 M, 2 F) of Mandarin Chinese (Fig 1A). Each tone category (e.g., high-flat, low-rising, low-dipping, and high-falling) was produced by each speaker in five syllable contexts (/bu/, /di/, /lu/, /ma/, and /mi/) for a total of 80 stimuli (20/category). The stimuli from two speakers (1 F, 1 M) were

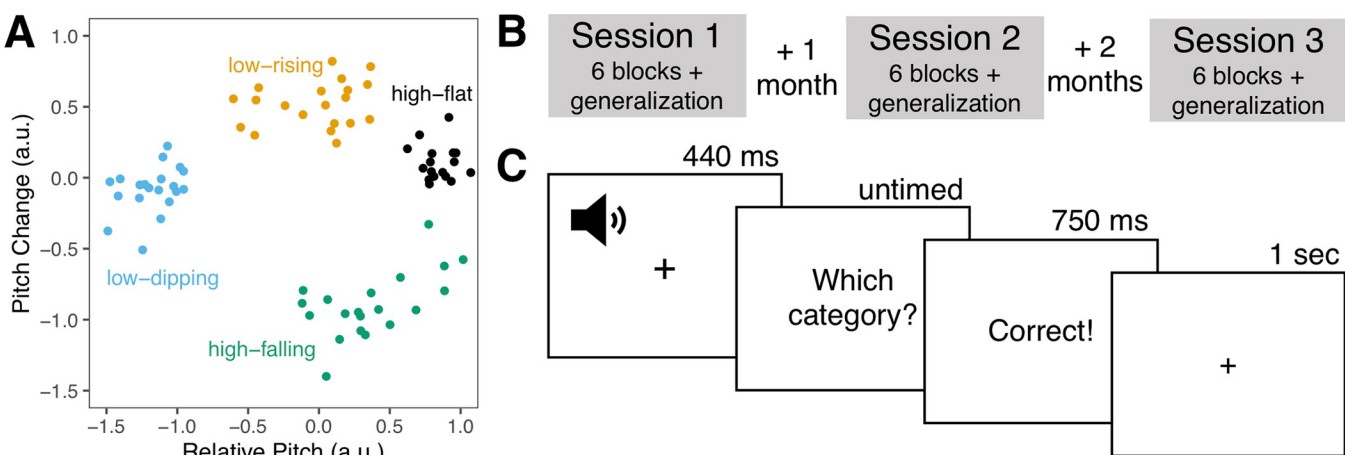

**Fig 1. Stimuli and procedure.** A. Two-dimensional representation of stimuli used during category learning and generalization with colors reflecting different tone categories. B. Session procedure. C. Task procedure.

used during the training blocks and the stimuli from the other speakers (1 F, 1 M) were withheld for the generalization block. The same 40 generalization stimuli were presented in the generalization block of each session and participants never received feedback about these stimuli. To reduce incidental differences in duration across categories, the stimuli were duration-normalized to 440 ms and RMS-amplitude normalized to 70 dB. The stimuli are shown in Fig 1A in a two-dimensional space (relative pitch, pitch change) that can be used to separate the stimuli into categories and is linked to neural representations of these categories [60, 61].

## Procedure

**Category learning.** Participants completed three separate sessions of category learning (Fig 1B). Sessions 1 and 2 were separated by one month. Sessions 2 and 3 were separated by two months. In each session, participants completed six blocks of an identical category learning task and an additional generalization block with different stimuli and no feedback. The stimuli were the same across sessions. Participants never received feedback about the generalization stimuli. At the beginning of the experiment, participants were told that they would be grouping sounds into different categories based on corrective feedback. They were not given any specific instructions about the stimuli or what might differentiate the categories from one another.

In the category learning task, there were six blocks of 40 trials each. In the generalization task, there was one block of 40 trials. Participants heard the 440 ms duration sound, followed by a prompt about the category identity ("Which category?") (Fig 1C). They pressed the 1, 2, 3, and 4 buttons on the keyboard to respond. Participants received trial-by-trial feedback in the category learning task where they were informed about whether their decision was 'Correct' or 'Incorrect.' The feedback was presented immediately for 750 ms. Participants did not receive feedback in the generalization task. In both tasks, there was an intertrial interval of 1 sec.

**Working memory capacity.** In the first session, participants first completed the category learning and generalization blocks and then completed an operation span task [62] as a measure of WM capacity. Participants were shown simple arithmetic problems and reported whether the presented solutions were correct or incorrect (e.g., (1 + 7) x 2 = 16) and were then shown a letter on the screen (e.g., A). A sequence of these arithmetic problems and letters from three to seven items in length made up a trial. After a full sequence was presented, participants were instructed to recall the letters presented in order. There were 15 trials. Participants' WM capacity was calculated based on the OSPAN score–the sum of the length of all correctly recalled spans. For example, if a participant correctly recalled a sequence of four letters (e.g., A, I, D, F), four points were added to their score. The minimum possible OSPAN score is 0 and the maximum possible OSPAN score is 75. We did not filter scores based on accuracy on the arithmetic problems [63] and participants were generally very accurate ($M$ = 85%, $SD$ = 14%; Fig B in S1 File).

## Drift diffusion modeling

We applied a variant of the DDMs developed in Paulon et al. [53]. The model estimates the evidence accumulation rate (i.e., drift) $\mu_{d,s}$ for each combination of decision response $d$ and stimulus category $s$ and decision thresholds (i.e., boundaries) $b_d$ for each decision response $d$. Additionally, the model also fits offset parameters $\delta_s$ for each stimulus category, which characterize the times taken by the actions that are not directly relevant to the actual decision-making processes (e.g., the time required to encode the $s$-th stimulus before evidence accumulation begins, to press a computer key, to record a response after a decision is reached, etc.). The model lets the parameters $\mu_{d,s}$ $b_d$ and $\delta_s$ to vary between participants, which accommodates the

substantial variability across participants. Importantly, the model also allows $\mu_{d,s}$ and $b_d$ to evolve smoothly over time (across training blocks), explaining the changes in the decision-making processes as the participants learn over time. We allowed the drift rates to vary across both stimulus category and response and assume that participants gather evidence towards each of the four possible response options at different rates depending on the true identity of the stimulus category. The decisions participants make in this task are tied directly to the sound category. Exemplars from within a sound category share characteristics and differ from exemplars from other sound categories. Due to the stimulus characteristics, participants may accumulate evidence at different rates for the different stimulus-response combinations. Boundaries only varied across response and different levels of response caution were not dependent on the true stimulus category.

The data were filtered to exclude very fast and very slow responses by removing the top and bottom 1% of all trials across all participants based on reaction time. The remaining data, comprising both correct and incorrect trials, were used to estimate the parameters. Since gradual improvements in making correct decisions characterize learning, in our discussions below, we emphasize heavily on inferring the drift rates associated with successful identification of the stimulus ($\mu_{d,s}$ for correct responses with $s = d$). Consideration of all responses does not change the overall results (see Fig D in S1 File, Table D in S1 File).

We adopted a Bayesian framework for these analyses, assigning priors to the parameters and relying on samples drawn from the posterior using a Markov chain Monte Carlo (MCMC) algorithm for estimation and inference. The algorithm was run for 6,000 iterations with the initial 2,000 iterations discarded as burn-in. The remaining samples were further thinned by an interval of 5 to reduce autocorrelation. MCMC diagnostics such as trace-plots of the parameters, Geweke test for stationarity of the chains, etc. indicated no convergence or mixing-related issues. Posterior predictive checks indicated good model fit. Finally, posterior means are reported as point estimates and pointwise credible intervals are used to assess uncertainty. For more details on the implementation of these models, see S1 File.

Data were visualized and analyzed using R, version 4.3.1 [64] and the following R packages: *tidyverse*, version 1.3.2, [65], *ggplot2*, version 3.4.3 [66], *ggthemes*, version 4.2.4 [67], *lddmm*, version 0.4.2 [68], *lme4*, version 1.1.34 [69], *lmerTest*, version 3.1.3 [70], *rstatix*, version 0.7.2 [71].

## Results

### Learning performance

On average, participants learned the Mandarin tone categories with substantial individual variability in performance (Fig 2A). For context, we also plot the reaction times (Fig 2B). We note that for visualization of performance across blocks, we grouped participants by their WM scores based on a median split (*Mdn* = 46), with values equal to or higher than the median defined as high WM and values lower than the median being defined as low WM. The analyses were conducted using raw OSPAN scores as a continuous variable with linear mixed effects models using the *lme4* package in R [69] and are also shown (Fig 2C).

We examined the extent to which WM capacity, indexed by the OSPAN score, was associated with performance in the category learning task. We used linear mixed effects models with session (as categorical variable), block, WM capacity, all possible interactions as fixed effects, participant (intercept) as a random effect, and average accuracy across a block as the continuous outcome variable. Session 1 was treated as the baseline session. Full results are presented in Table 2.

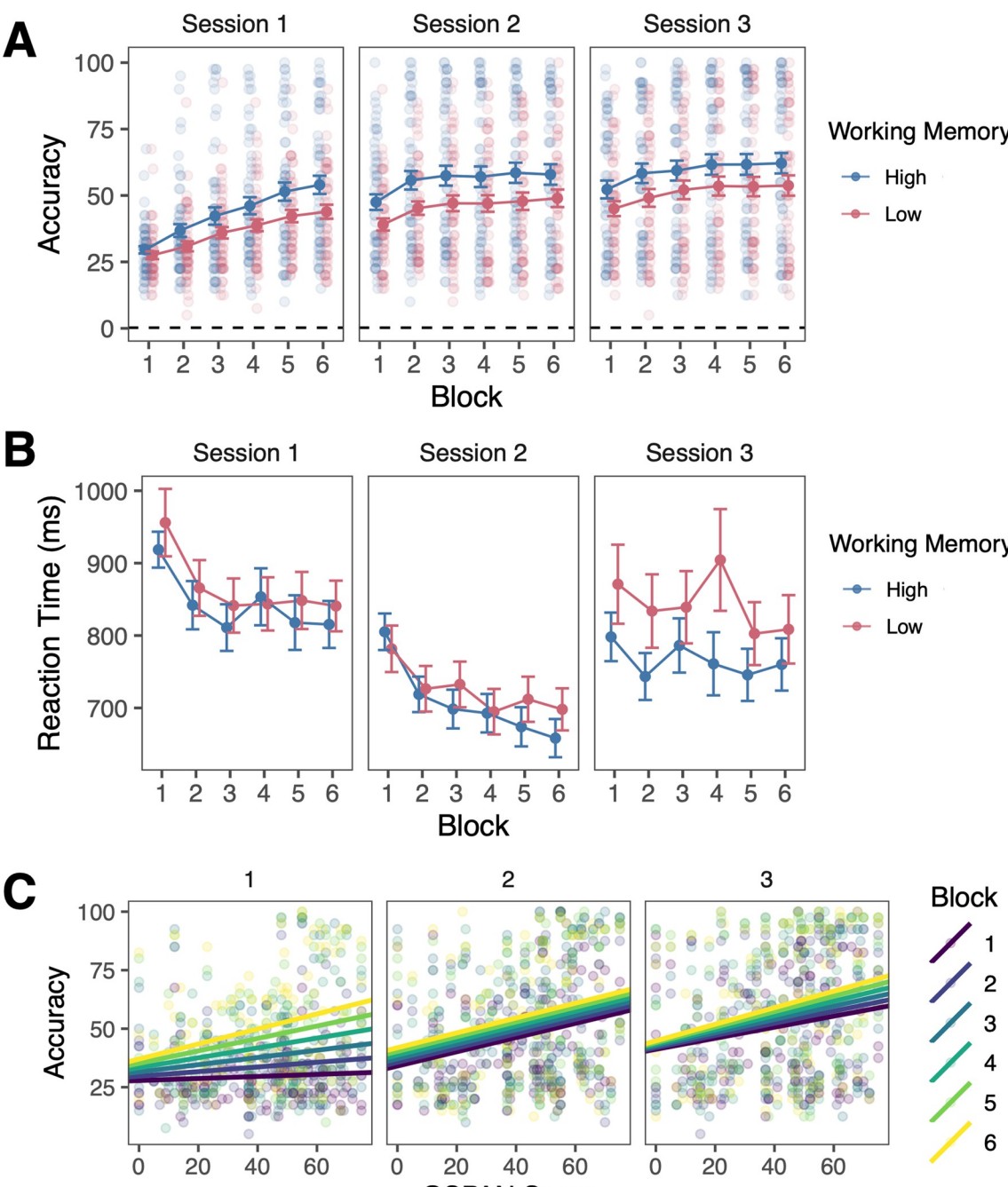

**Fig 2. Working memory and learning performance across all participants.** A. Accuracy and B. Reaction times after removing the shortest and longest 1% of responses. Error bars reflect *SEM*. For purposes of illustration, high and low working memory groups are defined based on a median split of working memory (OSPAN) scores. C. Relation between OSPAN score and proportion correct across blocks and sessions for all participants.

Overall, accuracy improved linearly across blocks in all sessions ($\beta_{Block}$ = 1.84, SE = 0.62, *p* = .0032; $\beta_{Block*Session2}$ = -0.28, SE = 0.88, *p* = .75; $\beta_{Block*Session3}$ = -1.12, SE = 0.88, *p* = .20) and improved marginally in session 2 from session 1 ($\beta_{Session2}$ = 6.51, SE = 3.43, *p* = .058) and significantly in session 3 from session 1 ($\beta_{Session3}$ = 14.5, SE = 3.43, *p* < .0001).

**Table 2. Summary of results on WM capacity and category learning performance.**

|  | β | SE | p |
|---|---|---|---|
| Intercept | 26.0 | 5.09 | < .0001 |
| OSPAN | -0.011 | 0.11 | .92 |
| Block | 1.84 | 0.62 | .0032 |
| Session 2 | 6.51 | 3.43 | .058 |
| Session 3 | 14.5 | 3.43 | < .0001 |
| OSPAN * Block | 0.055 | 0.013 | < .0001 |
| OSPAN * Session 2 | 0.31 | 0.074 | < .0001 |
| OSPAN * Session 3 | 0.22 | 0.074 | .0026 |
| Block * Session 2 | -0.28 | 0.88 | .75 |
| Block * Session 3 | -1.12 | 0.88 | .20 |
| OSPAN * Block * Session 2 | -0.053 | 0.019 | .0057 |
| OSPAN * Block * Session 3 | -0.032 | 0.019 | .095 |

β, estimate. *SE*, standard error of estimate. *p*, *p*-value. OSPAN, operation span score.

Collapsing across blocks, the relationship between WM score and accuracy was not significant in session 1 ($\beta_{OSPAN}$ = -0.011, SE = 0.11, $p$ = .92), but was significantly stronger in sessions 2 and 3 ($\beta_{OSPAN*Session2}$ = 0.31, SE = 0.074, $p$ < .0001; $\beta_{OSPAN*Session3}$ = 0.22, SE = 0.074, $p$ = .0026). Importantly, the relationship between WM score and accuracy interacted with both block and session. In session 1, there was a positive relationship between WM and accuracy that became stronger across blocks ($\beta_{OSPAN*Block}$ = 0.055, SE = 0.013, $p$ < .0001). A one unit increase in WM score was associated with an additional 0.055% increase in accuracy in each block. While in the first block, the relationship between WM score and accuracy was very weak (0.044%), by the final block, the relationship was clearly positive (0.32%). As a reminder, WM scores could range from 0 to 75, so even a relatively modest increase in WM score of 10 points would be associated with an additional increase in accuracy of 3.2% in the final block of session 1. A larger difference in WM score of 30 points would be associated with an additional increase in accuracy of 9.6% in this block.

One month later, in session 2, there was a positive relationship between WM and accuracy. While the relationship between WM and accuracy became stronger across blocks, the relative change was significantly smaller than in session 1 ($\beta_{OSPAN*Block*Session2}$ = -0.053, SE = 0.019, $p$ = .0057). In session 2, a one unit increase in WM score was associated with an additional 0.002% increase in accuracy in each block. Across blocks, the relationship between WM score and accuracy was similar to session 1 (range 0.30% - 0.31%).

Two months after session 2, in session 3, there was a positive relationship between WM and accuracy that became stronger across blocks in a way that was not significantly different from session 1 ($\beta_{OSPAN*Block*Session3}$ = -0.032, SE = 0.019, $p$ = .095). In session 3, a one unit increase in WM score was associated with an additional 0.023% increase in accuracy in each block. In the first block, the relationship between WM score and accuracy was 0.24% and by the final block, the relationship was similar to the final blocks of the other sessions (0.35%).

Taken together, we found that working memory ability was positively associated with speech category learning accuracy across training sessions, becoming relatively stronger across blocks in sessions 1 and 3 and was stable in session 2. While in the very initial stages of learning, WM score was not significantly related to accuracy (0.044% in first block of session 1), by the end of session 1 and persisting through the other sessions, WM score was positively related to accuracy (range 0.24% to 0.35%). The positive relationship between WM ability and

performance emerged within the first session and remained relatively stable throughout follow up sessions 2 and 3.

**Learners and non-learners.** Importantly, we also aimed to understand if the relationship between WM capacity and accuracy was present when considering only participants who learned the categories. We identified participants who performed at or below chance levels in the final block of session 3 (defined by 95% cumulative binomial probability, 40 trials, 0.25 probability of correct response = 25% +/- 10%) as 'non-learners' and those who performed better than chance as 'learners' (Fig 3A). Even though the non-learners were defined based on their accuracy in the final block of session 3, non-learners had significantly lower accuracy throughout all blocks (Bonferroni-corrected pairwise comparisons, $p < .001$), except for the first block of session 1 ($p = .078$). This underlines the necessity of considering learners separately from non-learners.

A total of 32% (34/107) of participants were classified as non-learners. WM scores for learners ($M = 44.1$) were marginally higher than non-learners ($M = 36.7$; $t(65.6) = 1.79$, $p = .078$, 95% CI [-0.84, 15.6]). This may indicate that individuals with lower WM may be more likely to be non-learners. It is important to note that we cannot completely rule out that non-learners with seemingly lower WM may have been generally disengaged in the experiment, leading to poorer performance in both the WM task and the category learning task. If this is the case, WM scores for these individuals may not reflect their true WM abilities. As post-hoc evidence that some participants may have been disengaged across tasks, we found that learners ($M = 90$%) performed better than non-learners ($M = 79$%) at identifying the arithmetic equations as correct or incorrect in the WM task ($t(44) = 3.49$, $p = .0011$, 95% CI [4.58, 17.1]; Fig B in S1 File). In the following analyses, we focus on the remaining 68% (73/107) of participants who are operationally defined as 'learners' in the category learning task. Because the accuracies of non-learners were within a low and highly restricted range by definition, we examined the relationship between WM score and accuracy for learners only.

To understand if the relationship between WM and category learning performance was present when examining learners only, we ran the same linear model analysis with learners only (Fig 3B; Table 3). Session 1 was treated as a baseline.

Of critical interest is whether WM score and accuracy were still positively related when examining only those who learned the categories. In session 1 ignoring block, the relationship between WM and accuracy was not significant ($\beta_{OSPAN} = -0.029$, SE = 0.10, $p = .76$). However, this relationship became stronger across blocks ($\beta_{OSPAN*Block} = 0.050$, SE = 0.015, $p = .00059$; $\beta_{OSPAN*Block*NonLearners} = -0.043$, SE = 0.026, $p = .11$). A one unit increase in WM score was associated with an additional 0.050% increase in accuracy in each block for learners. By the final block of session 1, a one unit increase in WM score was associated with a 0.27% increase in accuracy for learners.

In session 2, the relationship between WM and accuracy was positive and significantly stronger than session 1 ($\beta_{OSPAN*Session2} = 0.30$, SE = 0.081, $p = .00018$). Ignoring block, a one unit increase in WM was associated with an increase in accuracy of 0.27%. This relationship was relatively stable, becoming mildly weaker across blocks. The relationship between WM score and accuracy across blocks was significantly different from session 1 ($\beta_{OSPAN*Block*Session2} = -0.058$, SE = 0.021, $p = .0048$). A one unit increase in WM score was associated with an additional 0.008% decrease in accuracy in each block. By the final block of session 2, a one unit increase in WM score was associated with a 0.23% increase in accuracy for learners.

In session 3, the relationship between WM and accuracy was positive and significantly stronger than session 1 ($\beta_{OSPAN*Session3} = 0.18$, SE = 0.081, $p = .023$). Ignoring block, a one unit increase in WM was associated with an increase in accuracy of 0.15%. The relationship was relatively stable, becoming mildly stronger across blocks. The relationship between WM score

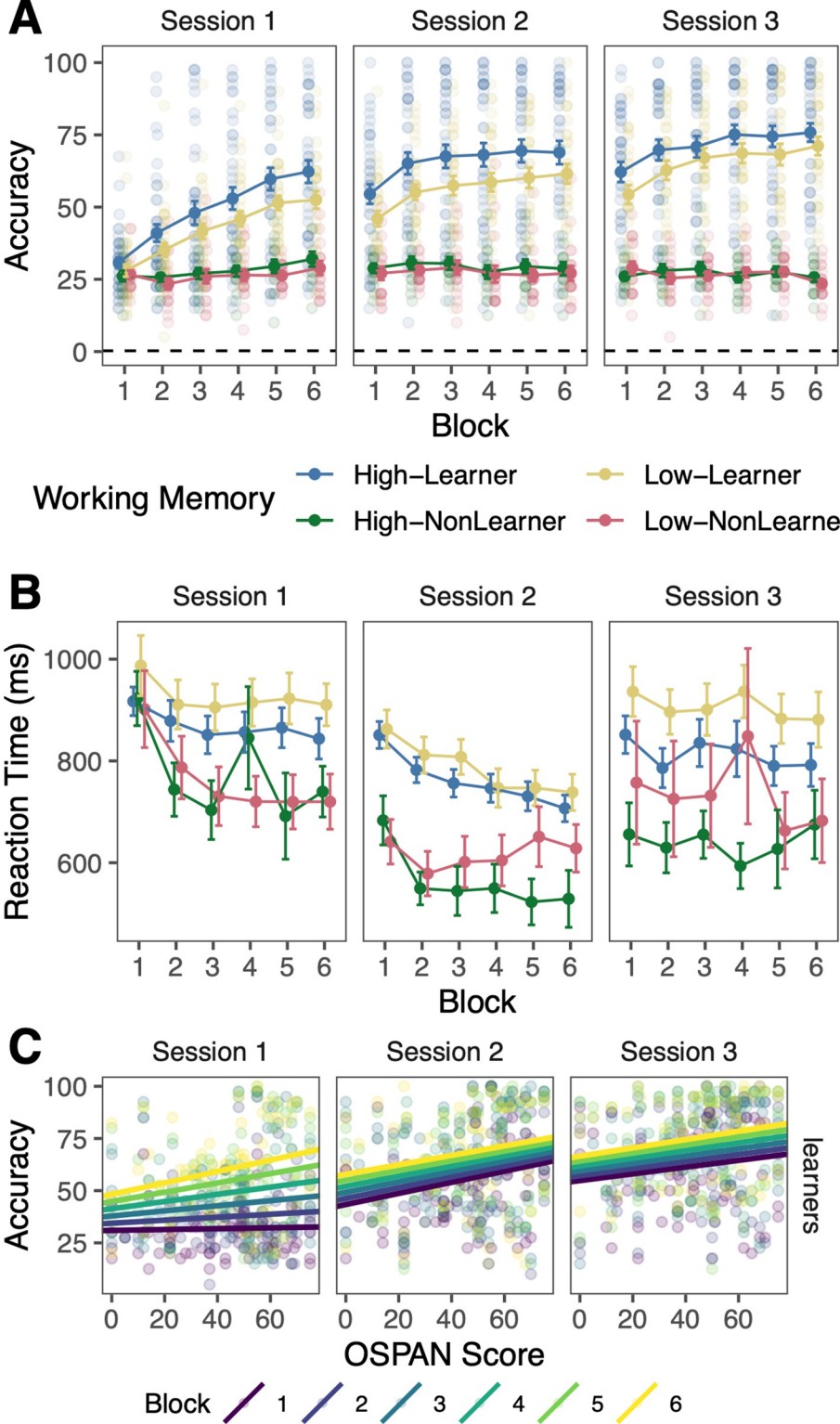

**Fig 3. Working memory and learning performance across learners and non-learners.** A. Accuracy and B. Reaction times after removing the shortest and longest 1% of responses. Error bars reflect *SEM*. For purposes of illustration, high and low working memory groups are defined based on a median split of working memory (OSPAN) scores. Groups are additionally separated into learners and non-learners based on session 3 block 6 accuracy and whether it was greater (learners) or less than (non-learners) chance performance. C. Relation between OSPAN score and proportion correct across blocks for learners only.

**Table 3. Summary of results on WM capacity and category learning performance across groups.**

|  | β | SE | p |
|---|---|---|---|
| Intercept | 27.5 | 4.94 | < .0001 |
| OSPAN | -0.029 | 0.10 | .78 |
| Block | 3.46 | 0.71 | < .0001 |
| Session 2 | 13.0 | 3.90 | .00090 |
| Session 3 | 25.0 | 3.90 | < .0001 |
| OSPAN * Block | 0.050 | 0.015 | .00059 |
| OSPAN * Session 2 | 0.30 | 0.081 | .00018 |
| OSPAN * Session 3 | 0.18 | 0.081 | .023 |
| Block * Session 2 | -0.56 | 1.00 | .58 |
| Block * Session 3 | -1.12 | 1.00 | .26 |
| OSPAN * Block * Session 2 | -0.058 | 0.021 | .0048 |
| OSPAN * Block * Session 3 | -0.043 | 0.021 | .039 |

β, estimate. *SE*, standard error of estimate. *p*, *p*-value. OSPAN, operation span score.

and accuracy across blocks was significantly different from session 1 ($\beta_{OSPAN*Block*Session3}$ = -0.043, SE = 0.021, *p* = .039). A one unit increase in WM score was associated with an additional 0.007% increase in accuracy in each block for learners. By the final block of session 3, a one unit increase in WM score was associated with a 0.20% increase in accuracy for learners.

Among learners only, higher WM ability was associated with better non-native speech category learning performance. This relationship emerged within the first session and was persistent across sessions 2 and 3 and, unsurprisingly, was slightly weaker than the relationship including all participants. The slope of the relationship between WM score and accuracy was 0.27% in the final block of session 1, 0.23% in the final block of session 2, and 0.20% in the final block of session 3.

## Maintenance of category knowledge over time

By examining learning across several sessions separated by one and two months, respectively, we can assess the maintenance of categorization performance and category knowledge over time. We assessed category knowledge maintenance by comparing adjacent training blocks that were either separated by no delay (i.e., blocks 5 and 6 of the same session) or a delay of one or two months (i.e., block 6 of one session and block 1 of the next session). Performance across these blocks and sessions for learners and non-learners separately is shown in Fig 4A. Because we are interested in how knowledge is retained over time, we focus our analyses only on learners.

Learners were somewhat able to maintain their category knowledge after a month or more of no additional training. Between sessions 1 and 2, accuracy fell an average of 7.2% (58.0% in block 6 to 50.9% in block 1) and between sessions 2 and 3, accuracy fell an average of 7.0% (65.6% in block 6 to 58.6% in block 1). In contrast, accuracy was relatively stable in the end of the sessions with accuracy increasing by 1.8% in session 1 (56.0% in block 5 to 57.8% in block 6) and by 0.3% in session 2 (65.2% in block 5 to 65.5% to block 6).

The ability to maintain category performance in adjacent blocks both with no delay (i.e., block 5 vs block 6) and after a one- or two-month delay (i.e., block 6 and block 1 of the next session) was unrelated to learners' WM capacity (Fig 4B, Table B in S1 File). We examined the percent difference between adjacent blocks across sessions using a linear mixed effects model with time (session 1 to 2 as baseline), delay (delay as baseline), WM score (OSPAN), and all

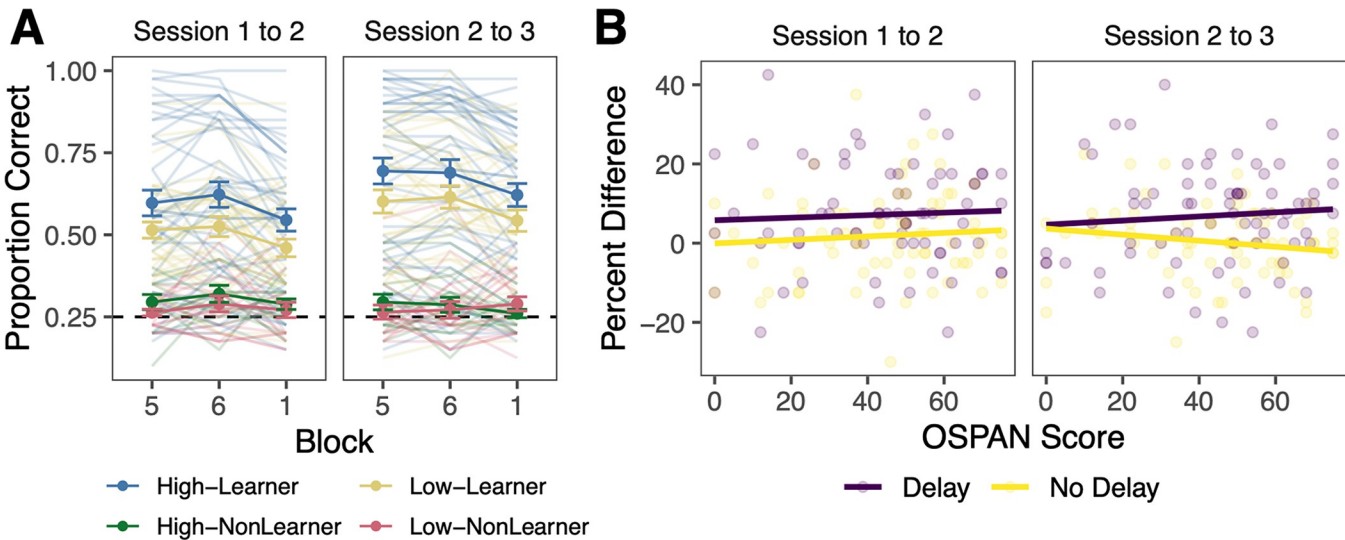

**Fig 4. Working memory and performance maintenance.** A. Error bars reflect *SEM*. For purposes of illustration, high and low working memory groups are defined based on a median split of working memory (OSPAN) scores. Groups are additionally separated into learners and non-learners based on session 3 block 6 accuracy and whether it was greater (learners) or less than (non-learners) chance performance. B. Relation between OSPAN score and percent difference from block 5 to 6 within a session (No Delay) and block 6 to block 1 (Delay) for learners only.

interactions as fixed effects and participant as a random effect. WM was unrelated to the retention of performance across sessions 1 to 2 ($\beta_{OSPAN}$ = 0.032, SE = 0.068, *p* = .64) and 2 and 3 ($\beta_{OSPAN*Sessions\ 2\ to\ 3}$ = 0.020, SE = 0.094, *p* = .83). The relationship between WM and retention did not depend on whether there was a delay of a month ($\beta_{OSPAN*Delay}$ = 0.012, SE = 0.094, *p* = .90) or two months ($\beta_{OSPAN*Delay*Sessions\ 2\ to\ 3}$ = -0.14, SE = 0.13, *p* = .29).

## Generalization to novel speakers

By examining how participants respond to new speakers about which they never receive feedback, we can assess the generalizability of their category knowledge. We first calculated a generalization score by subtracting the final training block accuracy from the test accuracy. Overall, learners were successful at generalizing their knowledge to the new speakers (Fig 5A). Once again, we focus our analyses on learners as there is no clear category knowledge for non-learners to generalize. We examined whether generalization performance across sessions was related to WM capacity by examining session (session 1 as baseline), WM score (OSPAN), and the interaction between session and WM score as fixed effects and participant as a random effect (Fig 5B, Table C in S1 File).

WM ability was not significantly related to learners' generalization ability in session 1 ($\beta_{OSPAN}$ = 0.080, SE = 0.055, *p* = .14). There were no significant differences in the relationship between WM and generalization accuracy in sessions 1 and 2 ($\beta_{OSPAN*Session2}$ = -0.049, SE = 0.076, *p* = .52) or sessions 1 and 3 ($\beta_{OSPAN*Session3}$ = -0.085, SE = 0.076, *p* = .26). Overall, these results demonstrate that, among learners, WM ability is not significantly related to the ability to generalize Mandarin tone category knowledge to novel speakers.

## Decision processes

We examined participants' decision processes based on the parameters from the drift diffusion models. We focus on the evidence accumulation rate (i.e., drift rate; Fig 6A) and decision threshold (i.e., boundary; Fig 6C) parameters. As these are Bayesian analyses, we interpret

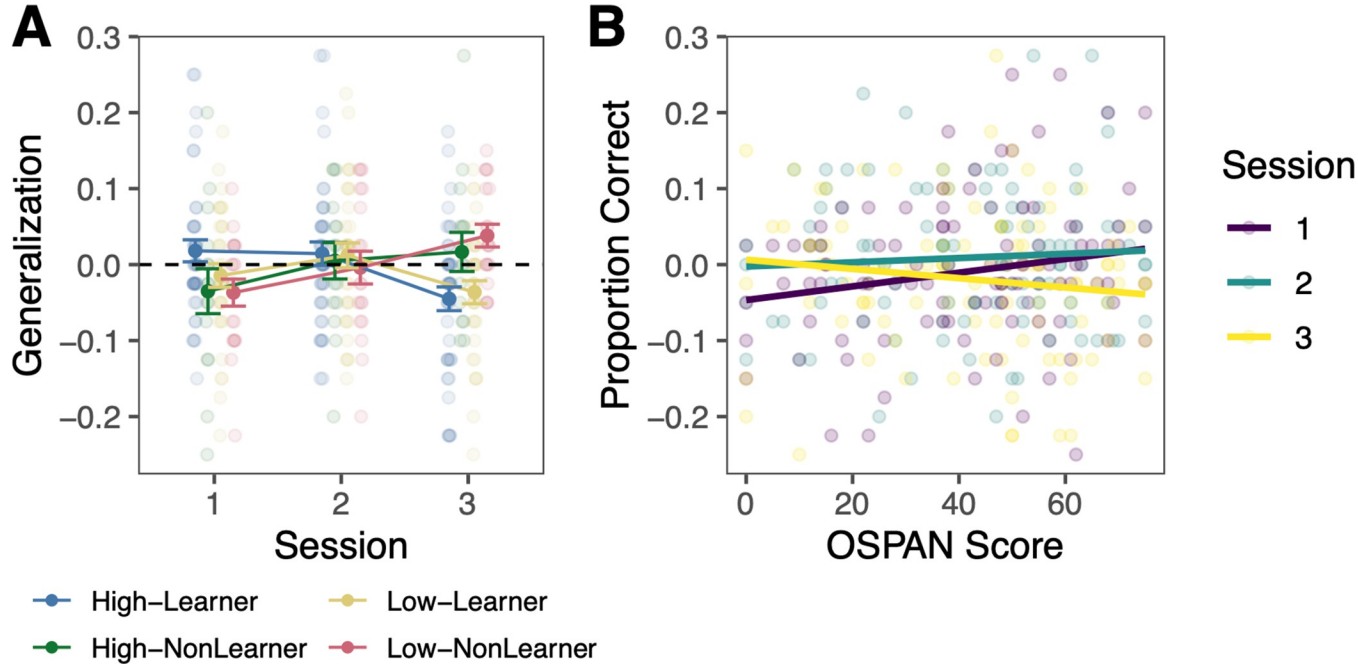

**Fig 5. Working memory and category generalization.** A. Error bars reflect *SEM*. For purposes of illustration, high and low working memory groups are defined based on a median split of working memory (OSPAN) scores. Groups are additionally separated into learners and non-learners based on session 3 block 6 accuracy and whether it was greater (learners) or less than (non-learners) chance performance. B. Relation between OSPAN score and generalization test score (mean generalization accuracy–mean block 6 accuracy) across sessions for learners only.

differences between groups where there is no overlap in the 95% credible intervals. We estimated the parameters for each individual and block, separately across sessions, with all subjects together (i.e., both learners and non-learners). As in prior work, we focus on the results for drift rates for accumulators where the stimulus category is the same as the response category (i.e., correct responses) [51]. This allows for examination of decision processes at play on trials where participants made correct responses. The overall pattern of results does not change when examining responses from all accumulators (Fig D in S1 File, Table D in S1 File).

First, we note the difference between learners and non-learners. Learners had higher evidence accumulation rates and higher decision thresholds than non-learners. In learners, the evidence accumulation rates increased over time, indicating that they became faster at accumulating evidence towards the correct decision. In contrast, the evidence accumulation rates in non-learners were low and flat throughout training, providing evidence of their general disengagement from the task. The decision thresholds were lower in non-learners than learners throughout the sessions deviating from one another after the very first block of training, indicating that non-learners needed less evidence to make their decision. This pattern may indicate that non-learners' decisions were based on optimizing speed rather than categorization accuracy.

Critically, our modeling approach enables estimation of the decision parameters at the *individual participant level*, allowing for examination of how these parameters relate to WM capacity. To understand how decision parameters differed based on WM in learners, we ran separate linear mixed effects models on the two parameters with block, session, WM score (OSPAN), and all interactions as fixed effects and participant as a random effect. Session 1 was treated as a baseline. Full results are shown in Tables 4 and 5. We focus on the results on the relationship between WM capacity and evidence accumulation rates and decision thresholds.

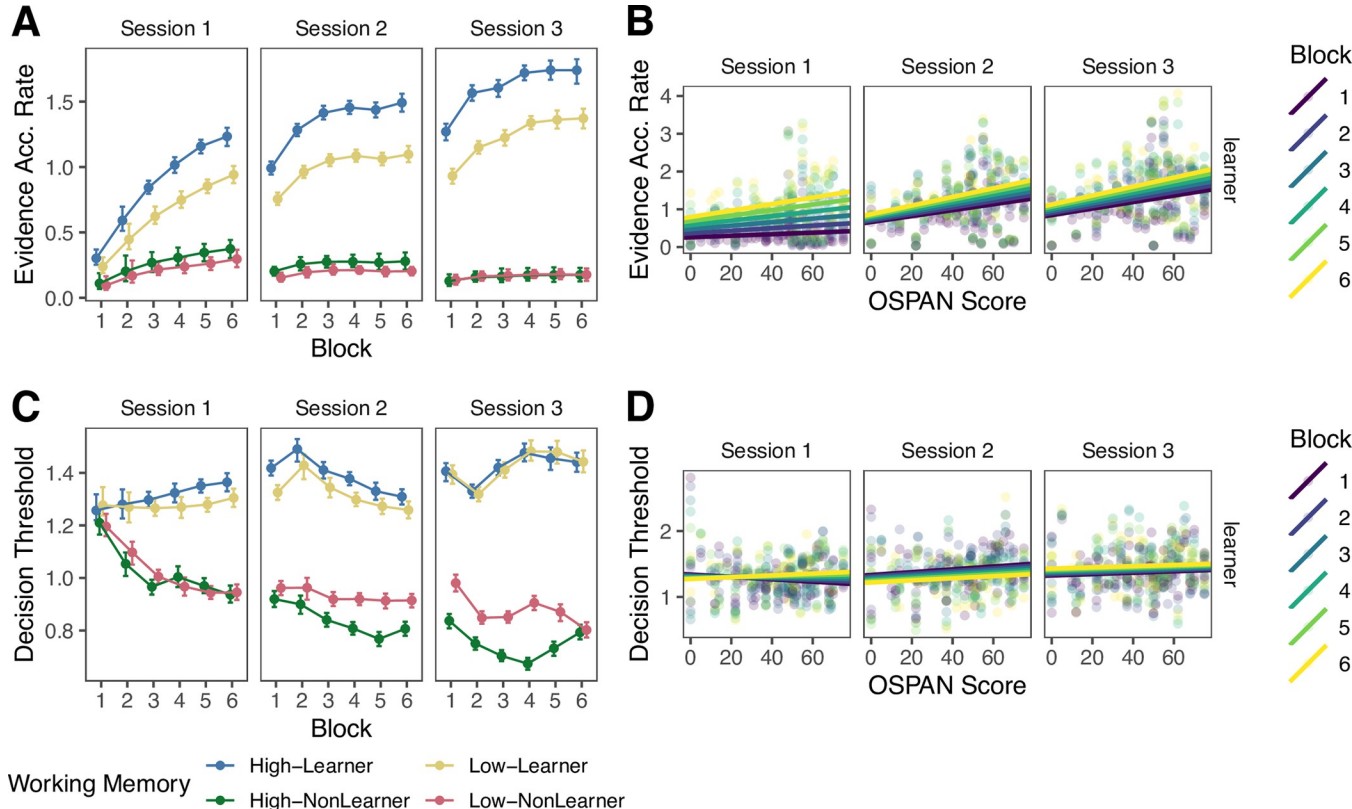

**Fig 6. Working memory and decision processes.** A and C: error bars reflect 95% credible intervals. For purposes of illustration, high and low working memory groups are defined based on a median split of working memory (OSPAN) scores. Groups are additionally separated into learners and non-learners based on session 3 block 6 accuracy and whether it was greater (learners) or less than (non-learners) chance performance. B and D: relation between OSPAN score and evidence accumulation rate and decision threshold for learners only.

Overall, learners with higher WM capacity accumulated evidence more quickly towards the correct decision in each session (Fig 6B). In session 1, there was not a significant relationship between WM and evidence accumulation rate ($\beta_{OSPAN}$ = 0.00075, SE = 0.0035, $p$ = .83).

**Table 4. Summary of results on WM capacity and evidence accumulation rate.**

|  | β | SE | p |
|---|---|---|---|
| Intercept | 0.15 | 0.17 | .38 |
| OSPAN | 0.00075 | 0.0035 | .83 |
| Block | 0.11 | 0.021 | < .0001 |
| Session 2 | 0.47 | 0.12 | < .0001 |
| Session 3 | 0.64 | 0.12 | < .0001 |
| OSPAN * Block | 0.0013 | 0.00044 | .0030 |
| OSPAN * Session 2 | 0.0064 | 0.0024 | .0084 |
| OSPAN * Session 3 | 0.0072 | 0.0024 | .0030 |
| Block * Session 2 | -0.066 | 0.030 | .029 |
| Block * Session 3 | -0.051 | 0.030 | .087 |
| OSPAN * Block * Session 2 | -0.00059 | 0.00062 | .34 |
| OSPAN * Block * Session 3 | -0.00065 | 0.00062 | .29 |

β, estimate. *SE*, standard error of estimate. *p*, *p*-value. OSPAN, operation span score.

**Table 5. Summary of results on WM capacity and decision threshold.**

| | β | SE | p |
|---|---|---|---|
| Intercept | 1.35 | 0.085 | < .0001 |
| OSPAN | -0.0024 | 0.0018 | .17 |
| Block | -0.014 | 0.013 | .31 |
| Session 2 | 0.0023 | 0.074 | .98 |
| Session 3 | -0.049 | 0.074 | .51 |
| OSPAN * Block | 0.00063 | 0.00028 | .022 |
| OSPAN * Session 2 | 0.0047 | 0.0015 | .0022 |
| OSPAN * Session 3 | 0.0035 | 0.0015 | .021 |
| Block * Session 2 | -0.0091 | 0.019 | .63 |
| Block * Session 3 | 0.035 | 0.019 | .068 |
| OSPAN * Block * Session 2 | -0.00074 | 0.00039 | .058 |
| OSPAN * Block * Session 3 | -0.00066 | 0.00039 | .090 |

β, estimate. *SE*, standard error of estimate. *p*, *p*-value. OSPAN, operation span score.

However, the relationship became significantly stronger across blocks ($\beta_{OSPAN*Block}$ = 0.0013, SE = 0.00044, *p* = .0030). A one unit increase in WM score was associated with an increase in evidence accumulation rate of 0.0021 units for learners in the first block of session 1 and 0.0086 units for learners in the final block of session 1.

The strength of the relationship between WM score and evidence accumulation rate also increased across sessions ($\beta_{OSPAN*Session2}$ = 0.0064, SE = 0.0024, *p* = .0084; $\beta_{OSPAN*Session3}$ = 0.0072, SE = 0.0024, *p* = .0030). In session 2, a one unit increase in WM score was associated with an increase in evidence accumulation rate of 0.0071 units for learners and this relationship was not significantly different across blocks ($\beta_{OSPAN*Block*Session2}$ = -0.00059 SE = 0.00062, *p* = .34). In session 3, a one unit increase in WM score was associated with an increase in evidence accumulation rate of 0.0079 units for learners and this relationship was not significantly different across blocks ($\beta_{OSPAN*Block*Session3}$ = -0.00065, SE = 0.00062, *p* = .29).

In contrast, there was no clear relationship between WM capacity and decision thresholds in any session (Fig 6B). In session 1, a one unit increase in WM score was associated with a non-significant decrease in threshold of 0.0024 units for learners ($\beta_{OSPAN}$ = -0.0024, SE = 0.0018, *p* = .17). The relationship between WM and threshold became slightly less negative across blocks in session 1 ($\beta_{OSPAN*Block}$ = 0.00063, SE = 0.00028, *p* = .022). A one unit increase in WM score was associated with a *decrease* in threshold of 0.0018 units for learners in the first block but an *increase* of 0.0014 units in the final block of session 1. Overall, in session 1, there was no clear relationship between WM score and decision threshold.

The relationship between WM and threshold differed in sessions 2 and 3 compared to session 1 ($\beta_{OSPAN*Session2}$ = 0.0047, SE = 0.0015, *p* = .0022; $\beta_{OSPAN*Session3}$ = 0.0035, SE = 0.0015, *p* = .021). However, this difference appears to stem from changing from a negligible negative relationship in session 1 to a negligible positive relationship in sessions 2 and 3. In session 2, one unit increase in WM score was associated with an increase in threshold of 0.0023 for learners, which did not significantly differ across blocks ($\beta_{OSPAN*Block*Session2}$ = -0.00074, SE = 0.00039, *p* = .058). In session 3, a one unit increase in WM score was associated with an increase in threshold of 0.0011 units for learners, which did not significantly differ across blocks ($\beta_{OSPAN*Block*Session3}$ = -0.00066, SE = 0.00039, *p* = .090). In sum, decision thresholds did not strongly relate to WM capacity in any session.

Overall, learners with higher WM capacity had *faster* evidence accumulation rates. The relationship began to emerge in the first session and was clearly present in the second and third sessions. In contrast, learners' decision thresholds did not depend on WM capacity. Together, these results indicate that WM capacity impacts specific elements of decision-making differently across the trajectory of learning.

## Discussion

We investigated non-native speech category learning in initial learning sessions and in two follow up sessions with one and two months between each session, respectively. We examined the extent to which WM capacity was related to initial and later learning sessions and in which ways (Fig 7). Considering all participants, higher WM was associated with better speech category learning across learning stages. Participants with higher WM may also have been more likely to learn the categories than participants with lower WM. When considering only individuals who performed at above-chance levels (i.e., learners), WM was associated with better performance by later blocks of initial acquisition (session 1) and in intermediate and later sessions (session 2–3) becoming somewhat weaker over time. WM ability was generally unrelated to maintenance of category knowledge over delays or generalization of category knowledge to new talkers. Finally, among learners, higher WM capacity was associated with faster evidence accumulation rates across learning sessions and was not associated with decision thresholds in any session.

### Learners and non-learners

Our results demonstrate that simply grouping all participants together does not tell a complete story because some participants clearly do not demonstrate learning, performing at chance levels even after extensive training. However, swiftly removing these non-learners as is common practice in the field [32–35] may obscure parts of the story as well. Participants who performed at or below chance levels at the end of three sessions of training were consistently poor performers across all blocks and sessions had marginally lower WM scores than learners. Importantly, it is possible that non-learners with lower WM scores may have been generally

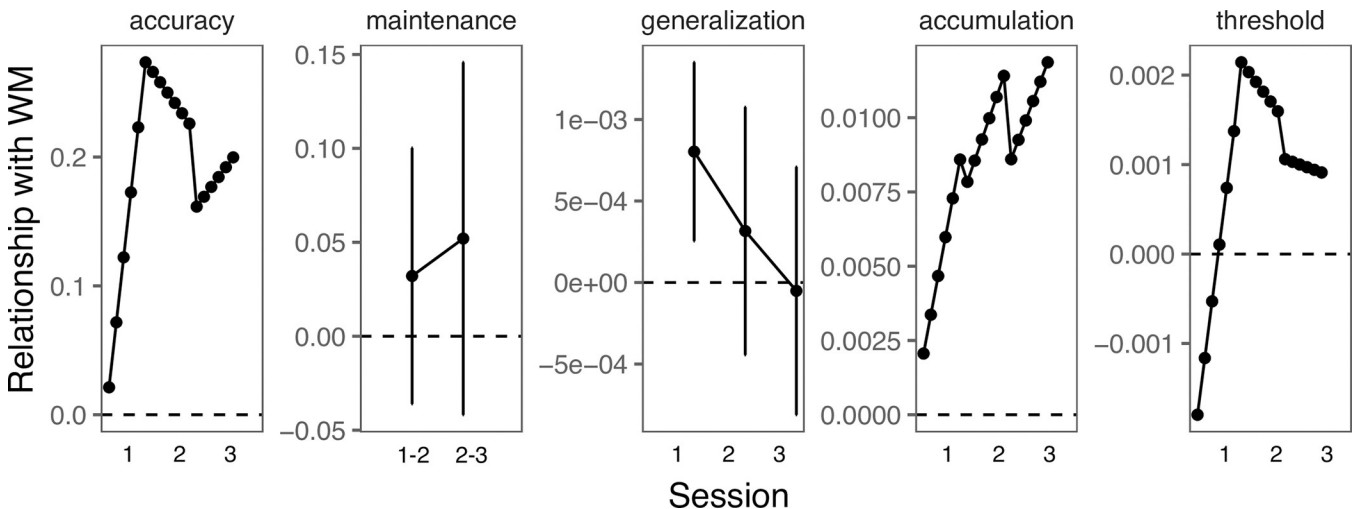

**Fig 7. Role of working memory in different stages of category learning.** Visualization of relationship between behavioral measures and working memory for learners based on the regression model coefficients. Error bars reflect *SEM*.

disengaged in the experiment, performing poorly across all measures (Fig B in S1 File). In support of the interpretation that non-learners were generally disengaged in the task, they had very low and flat evidence accumulation rates across learning, which may be indicative of general task disengagement [54, 72].

Regardless of WM ability, we found that a substantial number of participants (32%) were classified as non-learners. These individuals returned for three separate sessions of the same task that they were unable to consistently perform above chance levels. It is important to consider participants' goals and motivation for completing the task and compare this with experimenter-defined goals. Whereas we instructed them to respond as accurately as possible, their goal seemed to be to respond as quickly as possible regardless of accuracy evidenced by non-learners' much lower decision thresholds than learners. Decision thresholds (i.e., response caution) are related to the speed-accuracy tradeoff [57], with lower decision thresholds reflecting favoring speed over accuracy. As such, we interpret these low decision thresholds as a mark of these participants' disengagement in the category learning task. Importantly, favoring speed over accuracy is an adaptive strategy if your goal is not to *learn* the categories, but instead to complete the experiment as quickly as possible [73].

It is necessary to understand and adapt to the goals of our participants. This study was conducted using an online population, rather than a more typical convenience sample of college students leveraged in prior studies. This approach presents challenges, but also highlights that the goals and motivations to perform a simple experimental task may be different among a broader population than in student populations often examined in experimental psychology research.

It is important to understand how task disengagement is related to WM ability to understand potential interventions to improve learning. It is unclear if some non-learners want to learn, but they are unable to or if they are actively deciding to disengage from the task. Future work should include dynamic measures of task engagement, such as pupil dilation, to better understand how task engagement is related to WM and contributes to differences learning outcomes. If task disengagement is truly related to WM and we want to improve learning for individuals with lower WM, a first step should be ensuring that they are engaged in the task in the first place.

Together, these results highlight the importance of consideration of individual differences in learning. In particular, these results call for the need of special consideration of individuals who may be disengaged from the task. It is possible that a role that WM plays in learning is ensuring that resources are available for engagement in complex tasks.

## Initial learning and learning over time

The main goal of the current study was to understand the role of working memory in learning beyond initial acquisition. In line with prior work, we found that WM was positively related to learning by the end of the first session [7–12]. The benefit of higher WM in initial learning may stem from the ability to hold in mind many possible hypotheses which helps learners home in on the best one and use it faster and more efficiently [7, 12, 13]. Our results are in line with this prior work and suggest a role for WM in initial non-native speech category acquisition.

As a novel contribution, our results extend these findings and demonstrate that among participants who eventually learn the speech categories, WM was related to learning performance starting at the end of session 1 and persisting in sessions 2 and 3. This pattern of results conflicts with other work on speech category learning that demonstrates that given multiple days of training, there is no clear link between WM and performance [18, 19]. However, these prior studies trained participants on across days separated by very short delays, rather than delays of

over a month or more without additional training. Our results indicate that WM helps in initial acquisition of category knowledge, but individuals with lower WM may be able to 'catch up' given more time. Specifically, our results provide some preliminary evidence that the relationship between WM and non-native speech category learning may become weaker over time. Lower WM is not a sentence to poor learning forever. As long as participants remain engaged, they are able to learn.

This work also connects with prior investigations of learning from initial acquisition in novices to overtrained performance in experts in both language (e.g., [24]) and other perceptual contexts (e.g., [74]). While category representations start to emerge within a single session of training [30, 75], it is clear that further learning continues to shape representations and the networks supporting learning. For example, as individuals move from initial acquisition to highly experienced experts, there is a decrease in activation in sensory and frontal brain regions [76, 77], potentially reflecting increased neural efficiency with learning. Research from visual category learning demonstrates that similar neural networks support initial and well-learned categorization behavior, but that these networks become more coordinated with extensive practice [78]. Together, these results highlight the need to understand how learning and the cognitive abilities and processes that support categorization change from the very initial novice stages of learning to behavior in overtrained experts. This is particularly relevant for speech and language learning contexts, where expert or even genuinely stable levels of performance are unlikely to emerge in a single training session.

## Task difficulty and effort

We found that WM was consistently related to faster evidence accumulation among learners. These results are in line with prior work that demonstrates that evidence accumulation rates are linked to individual differences in WM [55, 56]. Faster evidence accumulation rates reflect higher motivation [54], faster mobilization of attentional resources [79], and lower task difficulty [80–84].

We then might interpret the persistently higher evidence accumulation rate in learners with higher WM as reflective of heightened motivation, rapid mobilization of available attentional resources, and perhaps perceived difficulty of the task. That is, even when accuracies were similar, learners with higher WM may have achieved that level of performance with lower perceived difficulty and perceived or exerted effort. Conversely, lower evidence accumulation rates observed in learners with lower WM may be associated with slower mobilization of motivational or attentional resources and more perceived difficulty in the task. Future research should clarify how WM relates to perceived difficulty and perceived and exerted effort during learning.

In summary, these results indicate that higher WM capacity is not a guarantee of better learning. Rather, it reflects better initial acquisition and general performance due to the ability to hold multiple hypotheses in mind and more rapid decision-making processes throughout learning. Lower WM also does not doom one to poor performance and, instead, lower WM may be linked to more time and resource-dependent decision processes which may be more effortful for the learner. Future work should address the perceived and exerted effort in learning and how this is related to WM.

## Limitations

We note that there was significant attrition across sessions. Whereas 195 individuals completed the first session, only 107 returned for both follow up sessions. This is a challenge for longitudinal designs using online samples but is a necessary challenge to overcome to understand learning beyond initial acquisition. While we considered non-learners who completed

all three sessions, it is also important to consider participants who failed to complete all parts of the experiment. In future work, it will be important to understand participants' reasons for returning or not returning to better understand what is motivating their performance in the task. Importantly, we found that WM did not differ based on how many sessions participants completed (Fig A-a in S1 File). This indicates that it was not just lower or higher WM individuals who failed to return for follow up sessions. There was also no difference in categorization accuracy based on the number of sessions participants completed. That is, within the same session, those who completed one, two, or all three sessions did not differ in their accuracy (Fig A-b in S1 File).

Another limitation of the current work is that we used a single measure of WM, measured at a single timepoint [85]. Specifically, we used an operation span measure based on ability to manipulate and remember a sequence of letters given a mathematical task interference. Operation span is extensively used and is a highly reliable measure of WM [85, 86]. Even still, one measure likely does not reflect the true complexity of WM. Further, because of the nature of the complex span task we used to assess WM capacity, it is possible that performance was influenced by some combination of WM and long-term memory [87]. As a result, the observed relationship between WM score and speech category learning performance may reflect the ability to hold onto and manipulate information in WM *as well as* retrieve exemplars or rules from long-term memory. However, it is important to note that measures that should theoretically be related to long-term memory or activation of exemplars stored in memory (e.g., maintenance, generalization) were not significantly related to WM score. Future studies should collect multiple measures of WM including visuospatial and auditory WM as well as measures of long-term memory to better understand how speech category learning relies on WM and long-term memory abilities.

Finally, participants learned four difficult categories with minimal feedback (e.g., "correct" or "incorrect"). Because this kind of feedback is ambiguous when the response is incorrect, it is possible that performance may have improved if we had provided full feedback (e.g., "correct, that was category 1"). However, prior work has demonstrated that Mandarin tone learning, as we examined here, is better with minimal feedback relative to full feedback [88]. Future studies will need to address the role of WM in learning with full and minimal feedback.

## Conclusion

We examined the relationship between WM and non-native speech category learning, maintenance of category knowledge across sessions, generalization to novel talkers, and decision processes involved in learning. The results demonstrate that higher WM is not a guarantee of learning, nor is lower WM a sentence to long-term learning difficulties. WM is one important ability in supervised category learning. Here, we highlight the need for a nuanced approach that considers the stage of learning and whether participants eventually learn. By leveraging a drift diffusion modeling approach and examining behavior from several angles over time, we conclude that WM may help learners by facilitating rapid category acquisition in initial stages and enhanced performance during subsequent stages of learning due to rapid evidence accumulation that may reduce the effort needed to learn. These results have important implications for developing interventions to improve learning in naturalistic language contexts and understanding what it means to be engaged in a task.

## Supporting information

**S1 File.**
(DOCX)

## Acknowledgments

All data, stimulus materials, and analysis code are publicly available at the Open Science Framework and can be accessed at https://doi.org/10.17605/OSF.IO/WDPYU. Data from the first session appeared in McHaney et al. (2021). Data from the second and third sessions have not appeared previously. Casey L. Roark is now at the University of New Hampshire, Department of Psychology. Jacie R. McHaney and Bharath Chandrasekaran are now at Northwestern University, Roxelyn and Richard Pepper Department of Communication Sciences and Disorders.

## Author Contributions

**Conceptualization:** Casey L. Roark, Jacie R. McHaney, Bharath Chandrasekaran.

**Data curation:** Casey L. Roark.

**Formal analysis:** Casey L. Roark, Giorgio Paulon, Giovanni Rebaudo, Abhra Sarkar.

**Funding acquisition:** Casey L. Roark, Abhra Sarkar, Bharath Chandrasekaran.

**Investigation:** Casey L. Roark, Jacie R. McHaney.

**Methodology:** Casey L. Roark.

**Project administration:** Jacie R. McHaney.

**Resources:** Bharath Chandrasekaran.

**Software:** Abhra Sarkar.

**Supervision:** Abhra Sarkar, Bharath Chandrasekaran.

**Validation:** Casey L. Roark.

**Visualization:** Casey L. Roark.

**Writing – original draft:** Casey L. Roark.

**Writing – review & editing:** Casey L. Roark, Jacie R. McHaney, Abhra Sarkar, Bharath Chandrasekaran.

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
