## [Decision Letter · Decision Letter 0]

2 Mar 2023

PONE-D-23-00404Individual differences in working memory impact the trajectory of non-native speech category learningPLOS ONE

Dear Dr. Roark,

Thank you for submitting your manuscript to PLOS ONE. I have sent your MS for evaluation by three experts on category learning, diffusion modeling, and working memory, and I have read your manuscript as well. First of all, I would like to thank the reviewers for their very insightful and constructive feedback, which you can find appended below. As you will see from their comments, they all appreciated your manuscript, the rigor of the methods, and the general approach. I agree with their assessment. The most substantial comment is related to the choices in implementing the drift-diffusion modeling, as well as the use of a linear regression with categorical predictors. Reviewer 1 would also appreciate if you could try to make more clear the main take home message of the manuscript communicating more clearly what is learned from the current analyses at the end. I believe that addressing these points will strengthen the MS and increase the impact of your work. Therefore I am inviting a Major Revision (which should also give you more time to address the raised points). Please consider all points raised by the reviewers one-by-one. At this stage, I do not have any further comments to add.

Just as final remark: thank you for the very comprehensive sharing of the data, materials and analysis code on the OSF. This is the correct spirit to allow for the complete reproducibility of your results.

We look forward to receiving your revised manuscript.

Kind regards,

Alessandra S. Souza, Ph.D.

Academic Editor

PLOS ONE

“This research was supported by the National Institute on Deafness and Other Communication Disorders [R01DC013315A1 to B.C., F32DC018979 to C.L.R., and T32DC011499 to K. Kandler and B. Yates (trainee: J.R.M.)] and the National Science Foundation [NSF-1953712 to B.C. & A.S.]. All data, stimulus materials, and analysis code are publicly available at the Open Science Framework and can be accessed at https://doi.org/10.17605/OSF.IO/WDPYU. Data from the first session appeared in McHaney et al. (2021). Data from the second and third sessions have not appeared previously. Corresponding authors: Casey L. Roark, casey.l.roark@gmail.com and Bharath Chandrasekaran, b.chandra@pitt.edu”

“This research was supported by the National Institute on Deafness and Other Communication Disorders [R01DC013315A1 to B.C., F32DC018979 to C.L.R., and T32DC011499 to K. Kandler and B. Yates (trainee: J.R.M.)] and the National Science Foundation [NSF-1953712 to B.C. & A.S.]. The funders had no role in study design, data collection and analysis, decision to publish, or preparation of the manuscript.”

3. We noted in your submission details that a portion of your manuscript may have been presented or published elsewhere. [DETAILS AS NEEDED] Please clarify whether this [conference proceeding or publication] was peer-reviewed and formally published. If this work was previously peer-reviewed and published, in the cover letter please provide the reason that this work does not constitute dual publication and should be included in the current manuscript.

Reviewers' comments:

Reviewer's Responses to Questions

**Comments to the Author**

1. Is the manuscript technically sound, and do the data support the conclusions?

Reviewer #1: Partly

Reviewer #2: Partly

Reviewer #3: Partly

2. Has the statistical analysis been performed appropriately and rigorously? 

Reviewer #1: I Don't Know

Reviewer #2: No

Reviewer #3: Yes

3. Have the authors made all data underlying the findings in their manuscript fully available?

Reviewer #1: Yes

Reviewer #2: Yes

Reviewer #3: Yes

4. Is the manuscript presented in an intelligible fashion and written in standard English?

Reviewer #1: Yes

Reviewer #2: Yes

Reviewer #3: Yes

5. Review Comments to the Author

Reviewer #1: Roark et al. present the results of research aimed at assessing the role of working memory in the learning and retention of categorical knowledge about non-native language stimuli. Participants completed a task online in which they learned Mandarin tone categories across three sessions, separated by months. The authors assessed the relationships between various aspects of task performance and participants' performance on an operation span task. In particular, they focused on relationships between OSPAN task performance and initial and later learning, maintenance of category knowledge across sessions, generalisation between trained and non-trained stimuli, and drift rate and threshold parameters from a multi-alternative diffusion model used to fit the data. Results showed positive relationships between OSPAN scores and categorisation accuracy in later learning (e.g., later blocks of session 1 and most blocks of subsequent sessions); however, when the sample was split into learners and non-learners, this relationship was not evident in either group. Maintenance of category knowledge across sessions and across the final blocks within a session was also unrelated to OSPAN scores, as was generalisation from training to test stimuli. Diffusion modelling showed positive relationships between OSPAN scores and both drift rate and threshold in later blocks of session 1, and in subsequent sessions, for learners. The authors conclude that greater WM capacity may aid category learning in language contexts, particularly in the early stages of acquisition; and that higher WM capacity allows for greater caution in decision-making owing owing to more rapid evidence accumulation.

I should mention at the outset that whereas my knowledge of the WM literature and evidence accumulation models of decision-making is fairly good, I am far from an expert when it comes to research into language learning. My comments thus largely focus on these former aspects of the work. I should also note that this makes it difficult for me to judge the theoretical value of the contribution this research makes to the language learning area.

I have four major comments concerning the paper (not presented in any particular order), and a few more minor ones.

Major comments:

1. My understanding is that theories of performance on complex span tasks (such as the OSPAN task) often assume a contribution from retrieval from LTM/episodic memory (e.g., Unsworth & Engle, 2007). If so, then I think a relevant question is, to what extent do the relationships between OSPAN performance and category learning shown here reflect the influence of LTM retrieval in both tasks? In other words, are the relationships evident in this work a result of the fact that both tasks to some extent tap how well people can retrieve stuff from LTM? Or is there something more to it than that? The meaning of the relationship depends on the answer to this question, and therefore I think it's something that would be worth addressing.

2. I was fairly confused by some aspects of the way the diffusion modelling was used in the paper. I had not come across this particular model before, but---if I understand it correctly---the basic idea seems straightforward: Assume Wald distributions for each response with densities that are multiplied by the product of the survival functions for the other responses. Nonetheless, there were two points that confused me: a) At one point (p. 24) you mention focusing on "correct trials only". I followed the reference used here (Roark et al., 2021, PB&R) to try to make sense of this, and unfortunately became even more confused. The source of my confusion is this: I don't understand how you can isolate a drift rate and threshold for correct trials only when all the responses must surely influence the estimated parameter values. The only thing I could think of was that you were only fitting the data from trials with correct responses; but this would lead to heavily biased parameter estimates (with the bias stronger when the proportion of errors increases). As such, I think you either need to clarify what it was you actually did, so that other readers don't end up as confused as I am; and/or re-do the modelling to fit all the data, rather than just those from correct trials. (On about my fifth read through the section, I think what you were actually doing here was fitting the model to all the data, but just reporting the parameters associated with the accumulator corresponding to the correct response. Right? This is more sensible than what I initially thought you were doing, but if this is correct, you really ought to clarify it so other readers don't go down the same garden path I did.) b) It seems to me that you use the estimated parameters for individual participants, obtained from the diffusion modelling, as the outcome variable in a regular linear regression with OSPAN score as the predictor. As I understand it though, the hierarchical structure of the diffusion model means these participant-level parameters are not independent, and therefore are not appropriate for use in the subsequent regression. Perhaps a better tack would be to include OSPAN scores as a predictor in the diffusion model itself.

3. Related to the previous point: If what you were doing with the diffusion modelling was indeed fitting all the data but just reporting parameters for the accumulator associated with the correct responses, I'd recommend also reporting the parameters for the other accumulators, at least in supplemental materials. In particular, when it comes to the effects of WM on thresholds (e.g., increasing with practice for higher-WM individuals), it's important to know whether this is unique to the accumulator for the correct response or consistent across all accumulators. If the effect is only evident on the accumulator for the correct response, this may indicate the model is not providing a good account of the data (since it is psychologically implausible that a participant could selectively adjust the threshold on the accumulator for the correct response before they know it to be the correct response). The solution to this would be to fix the accumulators to have the same thresholds.

4. I found it a little difficult to get a clear picture of the primary message you're intending the reader to take from the research. As I mentioned above, I am not very familiar with the language learning literature, so this may be an artefact of that unfamiliarity (e.g., findings with an importance that would be obvious to someone actively working in the area were not obvious to me). Even so, if you want the work to be more accessible to a general audience, it might be worth thinking about how you can better emphasise this message.

Minor comments:

p. 3, line 53: I think "temporal storage" should be "temporary storage".

pp. 3--4, lines 67--68: Really pedantic point here, but I think it's more precise to characterise performance as improving rather than increasing.

p. 4: In the discussion of the role of WM capacity in language learning (or elsewhere), I wonder whether it might be worthwhile bringing in recent work by Smalle (e.g., Smalle et al., 2021, 2022; see references below) suggesting that interfering with cognitive control mechanisms (an important part of WM) can improve statistical learning (an important part of language development)?

p. 8, lines 164--165: The phrasing is a bit awkward here.

p. 12, line 247: "were produced" should be "was produced".

p. 14, lines 290--291: There are a few differences between the study you cite here in support of not using an accuracy filter, and your own research (e.g., in-lab student vs. online sample) that make me wonder whether including an accurary threshold would be equally unnecessary in your case. I'd be interested to at least know how well participants performed in the arithmetic task (e.g., was the proportion of participants who did poorly similar to that in Ðokic et al.'s sample?). Actually I see from looking at the supplemental materials that you do provide arithmetic task performance data; perhaps good to mention this here.

p. 14, line 296: Did you consider allowing \\delta_s (i.e., the offset parameter) to vary across participants as well (and assess its relationship with OSPAN scores)? It seems plausible to me that part of learning stimuli in a novel language is developing the ability to more rapidly encode them, so that the decision (/categorisation) process receives good information more rapidly.

p. 15, line 307: What priors did you assign to the parameters?

p. 16, lines 338--346: Would a logistic model (e.g., via glmer) not be a better choice than a linear one here, given accuracy is restricted between 0 and 1 (/100)?

p. 17, lines 354--360: To enable the reader to better grasp the meaning of these effects, it might be useful to provide some information about the distribution of WM scores (e.g., to emphasise that a one-unit difference in WM scores is a fairly small increment); otherwise they may seem trivial.

p. 24, lines 490--491: When you say here that you "focus on the results for correct trials only", what exactly do you mean? Surely the model gives you parameter estimates that combine information from correct and error responses, no? In that case, how can you isolate the correct trials?

pp. 24--25, lines 509 on: Is there any reason for not just including WM capacity as a predictor in the diffusion modelling? Given the hierarchical nature of the diffusion model fitting, your individual participant parameter estimates are not independent, are they? Therefore the independence assumption of the regression analysis is violated.

p. 29, lines 620--625: The first thing that springs to my mind when I read this is Logan's instance theory of automatisation (e.g., Logan, 1988). If there is an initial stage of learning where a task is completed algorithmically, followed by expert responses resulting from LTM retrieval once enough instances are accumulated in memory, we would probably expect a fair bit of WM involvement in the first, but perhaps not so much in the second.

References:

Logan, G. D. (1988). Toward an instance theory of automatization. Psychological Review, 95(4), 492–527. https://doi.org/10.1037/0033-295X.95.4.492

Smalle, E. H. M., Muylle, M., Duyck, W., & Szmalec, A. (2021). Less is more: Depleting cognitive resources enhances language learning abilities in adults. Journal of Experimental Psychology: General, 150(12), 2423–2434. https://doi.org/10.1037/xge0001058

Smalle, E. H. M., Daikoku, T., Szmalec, A., Duyck, W., & Onen, R. M. (2022). Unlocking adults’ implicit statistical learning by cognitive depletion. Proceedings of the National Academy of Sciences of the United States of America, 119(2), 1–9. https://doi.org/10.1073/pnas.2026011119

Unsworth, N., & Engle, R. W. (2007). The nature of individual differences in working memory capacity: Active maintenance in primary memory and controlled search from secondary memory. Psychological Review, 114(1), 104–132. https://doi.org/10.1037/0033-295X.114.1.104

Reviewer #2: The authors examined the influence of working memory on category learning performance. Unlike the other studies, which either only studied the effect of WMC on immediate category learning or the effect after the categories were fully learned, the authors examined the continuum between the categories that were first formed to the time when the categories were fully acquired. Thus, the authors bridged the gap between the past studies and provided new insight into category learning. 

Overall, the experiment design was straightforward and logical. The flaws of doing an online study were carefully sidestepped and discussed. A large portion of chance-level participants might be dressed with a reward incentive. However, as a sufficient number of participants remained after excluding non-learners, running a new experiment might not be worth it. While the data is mostly sufficient to support the claim, the analysis is left to be desired.

There are two major concerns with the presented statistic:

1. The blocks were treated as categories instead of time series. This stops the author from looking at the IV change across blocks and can only look at the difference between specific blocks. Most importantly, supporting the "stable" performance claim is difficult without looking at the performance across the blocks. 

2. The statistic comes with vastly inflated alpha. The inflation might be the consequence of 1), as the interaction between the WMC and each block would have to be tested.

3. The authors claim the effect of WM changes over sessions (P.29, 619). However, the only statistic associated with the claim was the gradually (but not wholly) increasing p-value which happened to become insignificant at the later session (again, but not all blocks). An actual test of WMC effect across sessions would be a direct test of this claim. 

Drift diffusion model

The construction of the drift diffusion model also raised a question, why did the decision boundary depend on both the stimulus and the response? The authors used separate decision boundaries for four possible stimuli and four possible responses, i.e., 16 decision boundaries. Why? I can understand the difference in drift rate, as different stimuli would give different evidence for all possible responses. However, why would the decision boundaries change depending on the stimuli?

As the authors pointed out, the analysis only focused on the boundary leads to the correct response. Thus, implementing multiple boundaries based on the stimuli seems unnecessary and further explaination might be needed. If the extra boundaries were needed due to mathematical reasons. The authors needed to communicate it.

Some minor points:

P.7, Table 1: I think it might be dangerous to call the drift rate the "effieciency" of evidence accumulation. A stronger signal would also lead to a larger drift rate and is different from the "efficiency". 

P.7, Line 146: "A subset of these participants (107/195) return for the current study. This line doesn't fit into the previous sentences; maybe a misplacement?

P.10, Line 195: "The rate of evidence accumulation reflects the quality of information extracted from the stimulus, with faster rates reflecting a more efficient evidence accumulation process." This could also be on the increased efficiency of retrieving and comparing the examplers from memory. It would be difficult to identify whether the increase in drift rate was due to better information extraction or other categorization processes.

P.10, 200: Why can't the participants with higher WM to have the same decision threshold and respond faster without sacrificing the accuracy?

Reviewer #3: The authors investigate the relationship between working memory (WM; as measured by Operation Span) and learning Mandarin tone categories that were novel to the participants using a longitudinal, multi-session design. The authors examined both early learning as well as later retention/learning with follow-up sessions scheduled one month after the first and two months after the second. The authors also looked at whether there were differences for participants who successfully learned the tone categories vs. those who did not. For the latter, it appears that non-learners are generally disengaged from the task and may be seeking to minimize time spent in the experiment.

The key findings were that WM was positively associated with early learning, replicating several previous findings. Results for knowledge retention across sessions, however, showed no link, suggesting that WM is mainly involved in early, rather than later stages of learning. This is possibly due to the requirement of having to consider many different hypotheses early on in learning vs. only having to refine a single hypothesis (or a smaller set of viable ones) after performance has stabilized.

The authors also employ diffusion modeling to examine links between quality of evidence (drift rates, or accumulation rates) and response caution (decision thresholds). For learners, there was evidence that higher WM individuals showed not only more efficient evidence accumulation, but also greater response caution.

I found the manuscript to be very well written and exceptionally clear. The study was well-motivated, and the conclusions generally followed from the results of the analyses. My overall disposition is quite positive. I do have a few comments, mostly minor, that I would like to be addressed in a revision. I detail these below.

Major Comments

1 - Feedback was only provided as correct/error, but with a 4-category structure, this is not especially informative. Better to provide category feedback (i.e., information about response accuracy along with what the correct category was). This should be acknowledged as a limitation, as it may have contributed to poor performance leading to task disengagement. Because error feedback is highly ambiguous, it is that much more difficult to learn effectively.

2 - The racing diffusion model used here is set up in a way that differs from typical applications. The current model is configured in a way that is extremely flexible, raising questions about the viability of simpler, potentially more theoretically plausible, configurations. More seriously, the model configuration raises falsifiability concerns about the authors’ application of the model (see Jones & Dzhafarov, 2014).

My main concern is with the way decision thresholds are set. Based on my reading of the description on page 14, it seems that decision thresholds for the four categories are conditioned on the category of the stimulus (i.e., if a Category 1 stimulus is presented, the thresholds for the categories might be set to A, B, C, and D, but if a Category 2 stimulus is presented, they might be set at E, F, G, and H). If this is correct—apologies if I have misunderstood—the assumption is psychologically implausible, at it allows the system to adaptively configure itself in response to the stimulus category, but prior to identifying the stimulus category. A plausible configuration could still allow thresholds to differ across categories, but independent of the current stimulus.

Jones, M., & Dzhafarov, E. N. (2014). Unfalsifiability and mutual translatability of major modeling schemes for choice reaction time. Psychological Review, 121(1), 1–32. https://doi.org/10.1037/a0034190

3 - Related to this is whether a simpler model might provide a satisfactory account of the data. Given the unfamiliarity of the categories to the learners, is there an a priori reason to assume differences in decision thresholds? How well does a model with a single threshold common across all categories (and all stimuli) perform? The threshold(s) could still vary with learning, but testing a common threshold model against one that allows category-specific thresholds to vary would be prudent.

Minor Comments

4 – In Table 1, “enhanced learning and motivation” is listed as the hypothesized role of WM in later learning. Should this be “maintenance of knowledge”?

5 - Line 53 – Should “temporal” be “temporary”?

6 - Line 129 – The description of the two analyses considered could be explained more clearly in the parenthetical comment (i.e., learners only vs. all participants, including both learners and non-learners).

7 - Is the “non-decision time” offset, δs, allowed to vary across participants (and potentially learning)? If not, is there a reason why it is fixed across individuals/blocks?

8 - There have recently been a number of models that combine learning assumptions with diffusion decision mechanisms (e.g., Fontanesi et al., 2019, Miletic et al., 2021; Pedersen et al., 2017; Sewell et al., 2019). Do the authors think that their conclusions would be any different if one of these other models were used instead?

Fontanesi, L., Gluth, S., Spektor, M.S. et al. A reinforcement learning diffusion decision model for value-based decisions. Psychon Bull Rev 26, 1099–1121 (2019). https://doi.org/10.3758/s13423-018-1554-2

Steven Miletić, Russell J Boag, Anne C Trutti, Niek Stevenson, Birte U Forstmann, Andrew Heathcote (2021) A new model of decision processing in instrumental learning tasks eLife 10:e63055. https://doi.org/10.7554/eLife.63055

Pedersen, M.L., Frank, M.J. & Biele, G. The drift diffusion model as the choice rule in reinforcement learning. Psychon Bull Rev 24, 1234–1251 (2017). https://doi.org/10.3758/s13423-016-1199-y

Sewell, D.K., Jach, H.K., Boag, R.J. et al. Combining error-driven models of associative learning with evidence accumulation models of decision-making. Psychon Bull Rev 26, 868–893 (2019). https://doi.org/10.3758/s13423-019-01570-4

9 - Were fast responses filtered out of the data? Responses faster than 150-200 ms are usually regarded as anticipatory rather than stimulus-driven.

10 - Page 26 – The interpretation of decision thresholds in non-learners should be phrased more cautiously. Since most of the WM correlations are non-significant, there are concerns about false positives. Given that only one of the correlations outside of Session 1 was significant, I think the authors should be clear that the evidence that non-learners with higher WM have lower decision thresholds is relatively weak/inconsistent. That said, the more systematic trend in Session 1 is perhaps indicative of general task disengagement (or optimizing performance to minimize time in the experiment). Why this would only manifest in Session 1 is unclear, however, hence my general view that these results should be interpreted with greater caution.

The idea of non-learners trying to minimize time in the experiment is related to previous discussion along these lines by Hawkins, Brown, Steyvers, & Wagenmakers (2012).

Hawkins, G.E., Brown, S.D., Steyvers, M. et al. An optimal adjustment procedure to minimize experiment time in decisions with multiple alternatives. Psychon Bull Rev 19, 339–348 (2012). https://doi.org/10.3758/s13423-012-0216-z

11 - In the abstract, it is mentioned that WM is positively associated with generalization, but this doesn’t seem to be the case given the analyses reported on pp. 22-23.

6. PLOS authors have the option to publish the peer review history of their article (what does this mean?). If published, this will include your full peer review and any attached files.

Reviewer #1: No

Reviewer #2: **Yes: **Hsuan-Yu Lin

Reviewer #3: No

---

## [Author Response · Author response to Decision Letter 0]

29 Apr 2023

We thank the editor and three reviewers for consideration of our manuscript. Below, we briefly summarize the major changes we have made to the manuscript in response to reviewer comments. We then respond to each point raised by the reviewers with our responses beneath. We believe these changes significantly strengthen the manuscript. 

Summary: 

 Drift diffusion model details and analyses. To address comments from all three reviewers (R1: C2, 3, 11, 12, 15, 16, R2: C4, R3: C2, 3, 7, 8, 9), we have provided more information about the drift diffusion model parameters and analyses. We have also provided justification of our choice to allow boundaries to differ across stimulus and response categories and included a direct comparison of this full model with a sub-model that only allowed boundaries to differ across response categories. We demonstrated that the full model is psychologically plausible and provides a better statistical fit to the data than the sub-model. We believe these revisions substantially improve the manuscript and better contextualize the results. 

 Block as a continuous/categorical variable. Reviewer 2 (C1-3) raised valid concerns about our application of the regression models with block as a categorical variable. We agree with these concerns and have now re-ran the models using block as a continuous variable within sessions. We also directly compare performance across sessions. This enables us to better compare across blocks and sessions without an overly inflated alpha. 

 Takeaway messages. Reviewer 1 (C4) suggested that we edit the manuscript to make the takeaway messages clearer for readers. We have now drawn readers’ attention to the main takeaways throughout the manuscript and believe that this strengthens the manuscript as a whole. 

Reviewer comments and responses

Reviewer #1

Roark et al. present the results of research aimed at assessing the role of working memory in the learning and retention of categorical knowledge about non-native language stimuli. Participants completed a task online in which they learned Mandarin tone categories across three sessions, separated by months. The authors assessed the relationships between various aspects of task performance and participants' performance on an operation span task. In particular, they focused on relationships between OSPAN task performance and initial and later learning, maintenance of category knowledge across sessions, generalisation between trained and non-trained stimuli, and drift rate and threshold parameters from a multi-alternative diffusion model used to fit the data. Results showed positive relationships between OSPAN scores and categorisation accuracy in later learning (e.g., later blocks of session 1 and most blocks of subsequent sessions); however, when the sample was split into learners and non-learners, this relationship was not evident in either group. Maintenance of category knowledge across sessions and across the final blocks within a session was also unrelated to OSPAN scores, as was generalisation from training to test stimuli. Diffusion modelling showed positive relationships between OSPAN scores and both drift rate and threshold in later blocks of session 1, and in subsequent sessions, for learners. The authors conclude that greater WM capacity may aid category learning in language contexts, particularly in the early stages of acquisition; and that higher WM capacity allows for greater caution in decision-making owing owing to more rapid evidence accumulation.

I should mention at the outset that whereas my knowledge of the WM literature and evidence accumulation models of decision-making is fairly good, I am far from an expert when it comes to research into language learning. My comments thus largely focus on these former aspects of the work. I should also note that this makes it difficult for me to judge the theoretical value of the contribution this research makes to the language learning area.

I have four major comments concerning the paper (not presented in any particular order), and a few more minor ones.

Major comments:

R1.C1: 1. My understanding is that theories of performance on complex span tasks (such as the OSPAN task) often assume a contribution from retrieval from LTM/episodic memory (e.g., Unsworth & Engle, 2007). If so, then I think a relevant question is, to what extent do the relationships between OSPAN performance and category learning shown here reflect the influence of LTM retrieval in both tasks? In other words, are the relationships evident in this work a result of the fact that both tasks to some extent tap how well people can retrieve stuff from LTM? Or is there something more to it than that? The meaning of the relationship depends on the answer to this question, and therefore I think it's something that would be worth addressing.

Thank you for raising this point. We now directly address the potential involvement of LTM/episodic memory to performance in the OSPAN task in the interpretation of our results in the Discussion on pages 37-38 (lines 796-806): “Further, because of the nature of the complex span task we used to assess WM capacity, it is possible that performance was influenced by some combination of WM and long-term memory [88]. As a result, the observed relationship between WM score and speech category learning performance may reflect the ability to hold onto and manipulate information in WM as well as retrieve exemplars or rules from long-term memory. However, it is important to note that measures that should theoretically be related to long-term memory or activation of exemplars stored in memory (e.g., maintenance, generalization) were not significantly related to WM score. Future studies should collect multiple measures of WM including visuospatial and auditory WM as well as measures of long-term memory to better understand how speech category learning relies on WM and long-term memory abilities.”

R1.C2: 2. I was fairly confused by some aspects of the way the diffusion modelling was used in the paper. I had not come across this particular model before, but---if I understand it correctly---the basic idea seems straightforward: Assume Wald distributions for each response with densities that are multiplied by the product of the survival functions for the other responses. Nonetheless, there were two points that confused me: 

a) At one point (p. 24) you mention focusing on "correct trials only". I followed the reference used here (Roark et al., 2021, PB&R) to try to make sense of this, and unfortunately became even more confused. The source of my confusion is this: I don't understand how you can isolate a drift rate and threshold for correct trials only when all the responses must surely influence the estimated parameter values. The only thing I could think of was that you were only fitting the data from trials with correct responses; but this would lead to heavily biased parameter estimates (with the bias stronger when the proportion of errors increases). As such, I think you either need to clarify what it was you actually did, so that other readers don't end up as confused as I am; and/or re-do the modelling to fit all the data, rather than just those from correct trials. (On about my fifth read through the section, I think what you were actually doing here was fitting the model to all the data, but just reporting the parameters associated with the accumulator corresponding to the correct response. Right? This is more sensible than what I initially thought you were doing, but if this is correct, you really ought to clarify it so other readers don't go down the same garden path I did.) 

We indeed fit the model to data from all trials comprising both correct AND incorrect responses to estimate the parameters. However, since gradual improvements in making correct decisions characterize learning, in our discussion of the results, we emphasized heavily on inferring the parameters associated with successful identification of the input stimulus. We have edited the Methods section (page 16, lines 340-344) to explain these points more clearly: “The remaining data, comprising both correct and incorrect trials, were used to estimate the parameters. Since gradual improvements in making correct decisions characterize learning, in our discussions below, we emphasize heavily on inferring the parameters associated with successful identification of the stimulus, that is, µ_(d,s) and b_(d,s) for correct responses with s=d.”

b) It seems to me that you use the estimated parameters for individual participants, obtained from the diffusion modelling, as the outcome variable in a regular linear regression with OSPAN score as the predictor. As I understand it though, the hierarchical structure of the diffusion model means these participant-level parameters are not independent, and therefore are not appropriate for use in the subsequent regression. Perhaps a better tack would be to include OSPAN scores as a predictor in the diffusion model itself.

We used the estimated parameters for individual parameters from the diffusion modeling as the outcome variable. A single-stage estimation method with the OSPAN scores as a predictor in the diffusion model would be statistically more appropriate, we agree. However, while the multi-category multi-subject time-varying longitudinal drift-diffusion mixed models we have used here is a very sophisticated one, it is not straightforward to incorporate external continuous covariates in this model - we would need to carefully design such models, develop associated computational machinery, rigorously test them, etc. This is beyond the scope of the current article, but we do plan to pursue this statistics methodology problem as the topic of a separate project. We also note that multi-stage methods are quite widely used in the scientific literature when single-stage methods are not available. Based on our experience (in other contexts), multi-stage models, especially when they involve many latent variables (as with latent diffusion processes in drift-diffusion models), often produce more efficient and numerically stable inference. Finally, for this particular problem, we do not think a single-stage implementation will change the scientific conclusions we arrived at using our two-stage procedure. 

R1.C3: 3. Related to the previous point: If what you were doing with the diffusion modelling was indeed fitting all the data but just reporting parameters for the accumulator associated with the correct responses, I'd recommend also reporting the parameters for the other accumulators, at least in supplemental materials. In particular, when it comes to the effects of WM on thresholds (e.g., increasing with practice for higher-WM individuals), it's important to know whether this is unique to the accumulator for the correct response or consistent across all accumulators. If the effect is only evident on the accumulator for the correct response, this may indicate the model is not providing a good account of the data (since it is psychologically implausible that a participant could selectively adjust the threshold on the accumulator for the correct response before they know it to be the correct response). The solution to this would be to fix the accumulators to have the same thresholds.

We now report the parameters for all accumulators in the Supporting Information – specifically, we present the relationship between OSPAN score and evidence accumulation rate/decision threshold for the average of all stimuli and responses rather than just correct responses. We note that this does not change the overall pattern of results.

R1.C4: 4. I found it a little difficult to get a clear picture of the primary message you're intending the reader to take from the research. As I mentioned above, I am not very familiar with the language learning literature, so this may be an artefact of that unfamiliarity (e.g., findings with an importance that would be obvious to someone actively working in the area were not obvious to me). Even so, if you want the work to be more accessible to a general audience, it might be worth thinking about how you can better emphasise this message.

We thank the reviewer for this comment. We have now better summarized the primary takeaway messages for readers in the Abstract, Introduction, and Discussion. 

Minor comments:

R1.C5: p. 3, line 53: I think "temporal storage" should be "temporary storage".

We have changed this to “temporary storage”

R1.C6: pp. 3--4, lines 67--68: Really pedantic point here, but I think it's more precise to characterise performance as improving rather than increasing.

We have changed this to “improving”

R1.C7: p. 4: In the discussion of the role of WM capacity in language learning (or elsewhere), I wonder whether it might be worthwhile bringing in recent work by Smalle (e.g., Smalle et al., 2021, 2022; see references below) suggesting that interfering with cognitive control mechanisms (an important part of WM) can improve statistical learning (an important part of language development)?

Thank you for pointing us to these references. After reviewing these references, we have decided not to include them in our Discussion of our results because the methods are potentially too different from one another (e.g., unsupervised statistical learning of sequences vs. supervised learning of speech categories). We do not believe that we would be able to incorporate these references in the manuscript without providing substantial additional context and that is out of scope for the current article.

R1.C8: p. 8, lines 164--165: The phrasing is a bit awkward here.

We have updated this to: “The ability to accurately identify novel exemplars is a hallmark of categorization” (page 8, line 166-167)

R1.C9: p. 12, line 247: "were produced" should be "was produced".

We have updated this to “was produced”

R1.C10: p. 14, lines 290--291: There are a few differences between the study you cite here in support of not using an accuracy filter, and your own research (e.g., in-lab student vs. online sample) that make me wonder whether including an accurary threshold would be equally unnecessary in your case. I'd be interested to at least know how well participants performed in the arithmetic task (e.g., was the proportion of participants who did poorly similar to that in Ðokic et al.'s sample?). Actually I see from looking at the supplemental materials that you do provide arithmetic task performance data; perhaps good to mention this here.

We now directly report average performance on the arithmetic task in the main text (page 14, lines 298-300) and point interested readers to the Supporting Information for more information. 

R1.C11: p. 14, line 296: Did you consider allowing \\delta_s (i.e., the offset parameter) to vary across participants as well (and assess its relationship with OSPAN scores)? It seems plausible to me that part of learning stimuli in a novel language is developing the ability to more rapidly encode them, so that the decision (/categorisation) process receives good information more rapidly.

We did allow the offset parameter to also vary across participants. We have now clarified this explicitly on page 15, lines 308-310: “The model lets the parameters µ_(d,s), b_(d,s), and δ_s to vary between participants, which accommodates the substantial variability across participants.”

R1.C12: p. 15, line 307: What priors did you assign to the parameters?

The full details of our drift-diffusion model, including the choice of the priors, the resulting posterior, associated computational machinery for model fitting and posterior inference, simulated and real data illustrations, etc. can be found in Paulon et al (2021). A full discussion of these details is outside of the scope of this article. Following your comments, however, we now present a non-technical summary of the model in the Supporting Information.

R1.C13: p. 16, lines 338--346: Would a logistic model (e.g., via glmer) not be a better choice than a linear one here, given accuracy is restricted between 0 and 1 (/100)?

Even though accuracy is restricted between 0 and 1, the mean accuracy values are continuous and normally distributed and are therefore appropriate for a linear analysis. Because our foc

---

## [Decision Letter · Decision Letter 1]

21 Jun 2023

PONE-D-23-00404R1Individual differences in working memory impact the trajectory of non-native speech category learningPLOS ONE

Dear Dr. Roark,

 Thank you for submitting your manuscript to PLOS ONE. After careful consideration, we feel that it has merit but does not fully meet PLOS ONE’s publication criteria as it currently stands. Therefore, we invite you to submit a revised version of the manuscript that addresses the points raised during the review process.  Thank you for submitting your manuscript to PLOS ONE. I have sent your paper back to the original three reviewers of your prior submission. All reviewers agree that your paper is substantially improved. Yet, two of the reviewers feel that you have not addressed their concerns enough about the plausability of the model specified. This is a relatively big concern because model fit alone cannot be a basis to decide which model to use - when a model has several parameters, reasoning about the underlying model structure and meaningful implementation also needs to be used to constrain the model. Here, both reviewers feel that it makes no sense to allow response boundary to vary with the stimulus category. The reasoning for this is standard in the literature. Hence I agree with Reviewer 1 that you need to use a more constrained version, even if it slightly misfits the data. On that note, it would be important to present evidence of model fit. I think this should be presented as supplementary materials to not make the paper too long and cumbersome to read. Given that this is a major change in the results section, I am therefore inviting a major revision for the paper. Please include a point-by-point response to the reviewer's comments (note also that Reviewer 2 suggested that you may want to condense a bit the results section).

We look forward to receiving your revised manuscript.

Kind regards,

Alessandra S. Souza, Ph.D.

Academic Editor

PLOS ONE

Reviewers' comments:

Reviewer's Responses to Questions

**Comments to the Author**

1. If the authors have adequately addressed your comments raised in a previous round of review and you feel that this manuscript is now acceptable for publication, you may indicate that here to bypass the “Comments to the Author” section, enter your conflict of interest statement in the “Confidential to Editor” section, and submit your "Accept" recommendation.

Reviewer #1: (No Response)

Reviewer #2: All comments have been addressed

Reviewer #3: (No Response)

2. Is the manuscript technically sound, and do the data support the conclusions?

Reviewer #1: Partly

Reviewer #2: Yes

Reviewer #3: Partly

3. Has the statistical analysis been performed appropriately and rigorously? 

Reviewer #1: Yes

Reviewer #2: Yes

Reviewer #3: Yes

4. Have the authors made all data underlying the findings in their manuscript fully available?

Reviewer #1: Yes

Reviewer #2: Yes

Reviewer #3: Yes

5. Is the manuscript presented in an intelligible fashion and written in standard English?

Reviewer #1: Yes

Reviewer #2: Yes

Reviewer #3: Yes

6. Review Comments to the Author

Reviewer #1: The authors have effectively responded to most of my concerns, and I think the revised manuscript is an improvement on its predecessor in terms of its accessibility and readability. That said, I still have serious concerns regarding the psychological plausibility of a model of this type in which decision thresholds depend on knowledge of the stimuli being categorized, and the authors' response on this point has not convinced me otherwise. (I also read the paper by Paulon et al. very carefully to see whether I was missing something that had been described there, but did not come across any such thing.) A model that is psychologically implausible is problematic, irrespective of how well it might fit the data, because it is unclear how its parameters can be interpreted.

It is possible that people's decision boundaries really do change in a sort of cascaded manner, with processed low-level stimulus information feeding forward into the decision processes used for higher-level categorization. However, as far as I can see this would be an entirely different model to the one applied here.

I don't want to be totally un-constructive with my comments, so let me suggest three potential paths forward on this point. First, the authors could focus on the more restricted (but psychologically plausible) model instead, and revise the paper accordingly. Second, the authors could retain the current model, explicitly discuss its psychological implausibility, and explain in detail what sort of underlying processes might lead such a model to provide a better fit than the more restricted but theoretically sounder alternative. Third, the authors could do a bit of both: Cover the results from the restricted model AND the results from the flexible one, and discuss what it is that allows this model to fit the data better despite it being implausible, and what this might imply for the underlying processes. (Of course, I leave the possibility open that there are additional paths forward that I have neglected.)

In addition to this, I have just a few very minor, language-related comments:

p. 29, lines 610--612: This sentence is a little confusing. Do you mean that the increase across blocks was not significantly different from session 1? Or the rate itself?

p. 29, lines 619--620: "changed across blocks, sessions"---a word missing here?

p. 30, lines 634--635: I suggest changing "associated with an additional decrease in threshold" to "associated with a relative decrease in threshold" (or similar), since the baseline relationship for the session was positive.

p. 34, line 720: Who is the "them" in this sentence?

Reviewer #2: After the revision, the authors addressed all my previous concerns. Hence, I recommend accepting the manuscript.

The only minor gripe is that the statistic results became too long and cumbersome to read. Moving part of the results to the supplementary material might be better for readability. However, I would be okay with the paper staying the same.

Reviewer #3: The authors have addressed most of my comments from their initial submission. However, there remain two outstanding issues that have not been adequately addressed. These map onto my 2nd and 3rd points (R3.C2 and R3.C3, respectively) raised in my initial review.

Regarding R3.C2, I still have serious concerns about the decision to allow both drift rates and decision thresholds to vary as a function of stimulus-response combination. For drift rates, this is perfectly fine, and in keeping with conventional practice with regards to fitting data. For decision thresholds, however, this level of flexibility introduces a theoretical circularity that renders the model psychologically implausible.

It is fine for decision thresholds to vary as a function of response—participants may be more or less cautious about making particular responses. Indeed, this is how racing accumulator models (such as the racing diffusion model the authors use here) address response bias issues. Where things become problematic is when decision threshold is allowed to vary as a function of the stimulus. The core objection is this: If the outcome of the decision process is to identify the stimulus as belonging to a particular category, the system cannot be configured in a stimulus-specific manner because this requires foreknowledge of the outcome of the categorization process. Put another way, conditioning the threshold on the identity of the stimulus requires the system to have already categorized the stimulus for the purposes of threshold setting. In which case, there is no need for a subsequent evidence accumulation stage because the stimulus has already been categorized implicitly.

I appreciate that the authors have presented statistical support for allowing threshold to vary according to the stimulus, but this does not address the theoretical/psychological objection to configuring the model in this way. That is, the WAIC comparisons deal with a quality of fit issue, but they do not deal with the more substantive theoretical interpretation of the model.

This raises a point I had overlooked in my original review—there is no visualization of model fit. Is it the case that good fits can only be achieved by allowing the model to estimate thresholds in a stimulus-dependent way? Plotting either quantile-averaged data against quantile-averaged model predictions would be one way to check the correspondence between theory and data. Another way—perhaps more appropriate given the large sample nature of this study—would be to generate scatterplots showing predicted vs. observed RT quantiles for correct responses and errors as well as choice probabilities.

The other issue that I still think needs addressing, regarding R3.C3, extends on the above point. Are response-dependent thresholds required to achieve good fits to data? The authors correctly note in their original response that an accumulator framework does not require a common threshold setting for each accumulator, but my question was whether a common threshold value suffices to explain the data. It may well be the case that participants have response biases that lead them to be more reluctant/cautious about making some category responses over others, but demonstrating that this flexibility is needed is important for ensuring explanatory parsimony in terms of the preferred model.

7. PLOS authors have the option to publish the peer review history of their article (what does this mean?). If published, this will include your full peer review and any attached files.

Reviewer #1: No

Reviewer #2: No

Reviewer #3: No

---

## [Author Response · Author response to Decision Letter 1]

16 Nov 2023

We thank the editor and three reviewers for consideration of our revised manuscript. Below, we briefly summarize the additional changes we have made to the manuscript in response to reviewer comments. We then respond to each point raised by the reviewers with our responses directly underneath. 

Summary: 

 -Utilization of a more constrained version of the model (which allows drift rates to vary across stimulus and response categories, but boundaries only vary across response categories). Further, we arrived at this model by selecting among different versions of a more constrained model, which are now further detailed in the Supporting Information. 

 -Include more information about model fit in the Supporting Information.

 -Minor changes for readability and simplifying the Results section.

Editor

Thank you for submitting your manuscript to PLOS ONE. I have sent your paper back to the original three reviewers of your prior submission. All reviewers agree that your paper is substantially improved. Yet, two of the reviewers feel that you have not addressed their concerns enough about the plausability of the model specified. This is a relatively big concern because model fit alone cannot be a basis to decide which model to use - when a model has several parameters, reasoning about the underlying model structure and meaningful implementation also needs to be used to constrain the model. Here, both reviewers feel that it makes no sense to allow response boundary to vary with the stimulus category. The reasoning for this is standard in the literature. Hence I agree with Reviewer 1 that you need to use a more constrained version, even if it slightly misfits the data. On that note, it would be important to present evidence of model fit. I think this should be presented as supplementary materials to not make the paper too long and cumbersome to read.

RESPONSE:

In direct response to comments from the Editor as well as Reviewers 1 and 3, we have used a more constrained version of the model, which does not allow boundaries to vary across different stimulus categories. Additionally, we have added more evidence of model fit to the Supporting Information. 

We would also like to reiterate that the version of the model that allows both drift rates and boundaries to vary with the stimulus category is relatively novel to the field and is the basis of an NSF award to Dr. Sarkar and Dr. Chandrasekaran, but it does provide significantly better fit to the data. We understand that there is concern about the psychological plausibility of this version of the model (hence we have used a more constrained version here). While not fully elaborated therein, we can assure (three of the original authors are co-authors here as well) that the original decision to allow this flexibility in Paulon et al. (2021) was well-thought-out one, with significant deliberation regarding the biological plausibility, and not a mere oversight. 

Since the psychological plausibility of this model is outside of the scope of the current article, we plan to address this question in a separate future work. Our general rationale for the original model is as follows. (1) If we can assume (as is typical in conventional race models) that when an input stimulus category is presented, only the drift-diffusion processes corresponding to that specific input category (via drift rates varying by stimulus category) come into play in accumulating evidence in favor of different alternatives (while the other accumulators corresponding to other stimulus categories remain completely absent), we should be open to allowing the input category information influence other aspects of the underlying processes (including the boundaries) as well. (2) The most important timepoint to consider while interpreting the decision boundaries is at the time of decision, not at the beginning of or during the evidence accumulation process. Allowing the boundaries to be flexible based on incoming information from the stimulus reflects the ability to dynamically adjust behavior based on stimulus information. This ability may not be used in tasks and approaches that commonly use DDMs (e.g., detection, go-no go), but is arguably very important for perceptual categorization. (3) There is strong statistical evidence in favor of the fully flexible model (drift rates and boundaries allowed to vary over time and across responses and stimulus categories). If a reasonable argument for psychological plausibility can be made (see point 2 above), then we should take seriously the better statistical fit of this fully flexible model. 

Here (for the purpose of the response to reviewers only) we provide an assessment of model fit for four different models which only differ in terms of how they treat the boundaries. They are 

 flexible: the original model of Paulon, et al. (2021) that allows the boundaries b_(d,s)^((i)) (t) to vary with both the stimulus s and the response d as well as with time t; 

 fixed: a sub-model that allows the boundaries b_d^((i)) (t) to vary with the response d as and with time t but not with the stimulus s;

 constant: a sub-model that allows the boundaries b_(d,s)^((i)) (t) to vary with both the stimulus s and the response d but not with time t;

 fixed-constant: a sub-model that allows the boundaries b_d^((i)) to vary with the response d but not with stimulus s, nor with time t.

All models accommodate subject heterogeneity by allowing the boundaries to vary with the subject index i. As anticipated, the original ‘flexible’ model provides the best model fit, followed by ‘fixed’, then ‘constant’, and finally the most restrictive ‘fixed-constant’.

Table R1

Model fit assessed by -2*WAIC for the original model of Paulon et al. (2021) and three different sub-models. The smaller the reported model assessment value, the better. 

 Session 1 Session 2 Session 3

Model with b_(d,s)^((i)) (t) (flexible) 92691.86 82792.24 76261.58

Sub-Model with b_d^((i)) (t) (fixed) 93409.11 83795.52 77647.32

Sub-Model with b_(d,s)^((i)) (constant) 95203.27 84444.15 78203.95

Sub-Model with b_d^((i))(fixed-constant) 95697.74 85671.44 79635.96

Nevertheless, in the end, in the revised version of the current article and the Supporting Information document, we have decided to follow Reviewer 1’s first suggestion (see R1.C1 below) and used the more constrained ‘fixed’ version of the model, which does not allow boundaries to vary across different stimulus categories. We have edited the manuscript accordingly and the results for decision threshold are changed from previous versions. The results for the evidence accumulation rate are nearly identical to previous versions. The results overall have thus not changed substantially and the major conclusions very much remain the same. We hope this comprehensive revision that aligns with the reviewer perspectives addresses the concerns. 

Reviewer #1

R1.C1: The authors have effectively responded to most of my concerns, and I think the revised manuscript is an improvement on its predecessor in terms of its accessibility and readability. That said, I still have serious concerns regarding the psychological plausibility of a model of this type in which decision thresholds depend on knowledge of the stimuli being categorized, and the authors' response on this point has not convinced me otherwise. (I also read the paper by Paulon et al. very carefully to see whether I was missing something that had been described there, but did not come across any such thing.) A model that is psychologically implausible is problematic, irrespective of how well it might fit the data, because it is unclear how its parameters can be interpreted.

It is possible that people's decision boundaries really do change in a sort of cascaded manner, with processed low-level stimulus information feeding forward into the decision processes used for higher-level categorization. However, as far as I can see this would be an entirely different model to the one applied here.

I don't want to be totally un-constructive with my comments, so let me suggest three potential paths forward on this point. First, the authors could focus on the more restricted (but psychologically plausible) model instead, and revise the paper accordingly. Second, the authors could retain the current model, explicitly discuss its psychological implausibility, and explain in detail what sort of underlying processes might lead such a model to provide a better fit than the more restricted but theoretically sounder alternative. Third, the authors could do a bit of both: Cover the results from the restricted model AND the results from the flexible one, and discuss what it is that allows this model to fit the data better despite it being implausible, and what this might imply for the underlying processes. (Of course, I leave the possibility open that there are additional paths forward that I have neglected.)

RESPONSE:

In direct response to this comment (as well as similar comments made by Reviewer 3 and the Editor), we have used a more constrained version of the model, which does not allow boundaries to vary across different stimulus categories (Reviewer 1’s first suggestion here). Please see our detailed response to the Editor above for additional information. 

In addition to this, I have just a few very minor, language-related comments:

R1.C2: p. 29, lines 610--612: This sentence is a little confusing. Do you mean that the increase across blocks was not significantly different from session 1? Or the rate itself?

RESPONSE:

Based on overall edits to the results, this sentence is no longer in the manuscript. 

R1.C3: p. 29, lines 619--620: "changed across blocks, sessions"---a word missing here?

RESPONSE:

Based on overall edits to the results, this sentence is no longer in the manuscript.

R1.C4: p. 30, lines 634--635: I suggest changing "associated with an additional decrease in threshold" to "associated with a relative decrease in threshold" (or similar), since the baseline relationship for the session was positive.

RESPONSE:

Based on overall edits to the results, this sentence is no longer in the manuscript.

R1.C5: p. 34, line 720: Who is the "them" in this sentence?

RESPONSE:

We have changed this to clarify: “helps learners”

Reviewer #2

R2.C1: After the revision, the authors addressed all my previous concerns. Hence, I recommend accepting the manuscript.

The only minor gripe is that the statistic results became too long and cumbersome to read. Moving part of the results to the supplementary material might be better for readability. However, I would be okay with the paper staying the same.

RESPONSE:

Based on overall edits to the results to accommodate the more constrained model, the results are much shorter. We believe that these changes enhance readability. 

Reviewer #3

The authors have addressed most of my comments from their initial submission. However, there remain two outstanding issues that have not been adequately addressed. These map onto my 2nd and 3rd points (R3.C2 and R3.C3, respectively) raised in my initial review.

R3.C1: Regarding R3.C2, I still have serious concerns about the decision to allow both drift rates and decision thresholds to vary as a function of stimulus-response combination. For drift rates, this is perfectly fine, and in keeping with conventional practice with regards to fitting data. For decision thresholds, however, this level of flexibility introduces a theoretical circularity that renders the model psychologically implausible.

It is fine for decision thresholds to vary as a function of response—participants may be more or less cautious about making particular responses. Indeed, this is how racing accumulator models (such as the racing diffusion model the authors use here) address response bias issues. Where things become problematic is when decision threshold is allowed to vary as a function of the stimulus. The core objection is this: If the outcome of the decision process is to identify the stimulus as belonging to a particular category, the system cannot be configured in a stimulus-specific manner because this requires foreknowledge of the outcome of the categorization process. Put another way, conditioning the threshold on the identity of the stimulus requires the system to have already categorized the stimulus for the purposes of threshold setting. In which case, there is no need for a subsequent evidence accumulation stage because the stimulus has already been categorized implicitly.

I appreciate that the authors have presented statistical support for allowing threshold to vary according to the stimulus, but this does not address the theoretical/psychological objection to configuring the model in this way. That is, the WAIC comparisons deal with a quality of fit issue, but they do not deal with the more substantive theoretical interpretation of the model.

This raises a point I had overlooked in my original review—there is no visualization of model fit. Is it the case that good fits can only be achieved by allowing the model to estimate thresholds in a stimulus-dependent way? Plotting either quantile-averaged data against quantile-averaged model predictions would be one way to check the correspondence between theory and data. Another way—perhaps more appropriate given the large sample nature of this study—would be to generate scatterplots showing predicted vs. observed RT quantiles for correct responses and errors as well as choice probabilities.

RESPONSE: 

In direct response to the first point in this comment (as well as similar comments made by Reviewer 1 and the Editor), we have used a more constrained version of the model, which does not allow boundaries to vary across different stimulus categories (Reviewer 1’s first suggestion here). Please see our detailed response to the Editor above for additional information. 

RESPONSE:

To the second point regarding the visualization of model fit, we have now plotted the predicted versus observed reaction times (and responses) to the Supporting Information (Figure S3). 

R3.C2: The other issue that I still think needs addressing, regarding R3.C3, extends on the above point. Are response-dependent thresholds required to achieve good fits to data? The authors correctly note in their original response that an accumulator framework does not require a common threshold setting for each accumulator, but my question was whether a common threshold value suffices to explain the data. It may well be the case that participants have response biases that lead them to be more reluctant/cautious about making some category responses over others, but demonstrating that this flexibility is needed is important for ensuring explanatory parsimony in terms of the preferred model. 

RESPONSE:

Please see our detailed response to the Editor above.

---

## [Decision Letter · Decision Letter 2]

9 Jan 2024

PONE-D-23-00404R2Individual differences in working memory impact the trajectory of non-native speech category learningPLOS ONE

Dear Dr. Roark,

Thank you for submitting your manuscript to PLOS ONE. I sent your paper back to the more critical reviewers from the prior submission. They all believe you have done a great job in revising your paper. They have only made very small suggestions of clarifications for the text, which I am inviting you to correct and resubmit for final acceptance of the paper (see comments below). Thank you for considering PLOS ONE as the outlet for your work.

We look forward to receiving your revised manuscript.

Kind regards,

Alessandra S. Souza, Ph.D.

Academic Editor

PLOS ONE

Journal Requirements:

Reviewers' comments:

Reviewer's Responses to Questions

**Comments to the Author**

1. If the authors have adequately addressed your comments raised in a previous round of review and you feel that this manuscript is now acceptable for publication, you may indicate that here to bypass the “Comments to the Author” section, enter your conflict of interest statement in the “Confidential to Editor” section, and submit your "Accept" recommendation.

Reviewer #1: (No Response)

Reviewer #3: (No Response)

2. Is the manuscript technically sound, and do the data support the conclusions?

Reviewer #1: Yes

Reviewer #3: Yes

3. Has the statistical analysis been performed appropriately and rigorously? 

Reviewer #1: Yes

Reviewer #3: Yes

4. Have the authors made all data underlying the findings in their manuscript fully available?

Reviewer #1: Yes

Reviewer #3: Yes

5. Is the manuscript presented in an intelligible fashion and written in standard English?

Reviewer #1: Yes

Reviewer #3: Yes

6. Review Comments to the Author

Reviewer #1: The authors' current version of the manuscript has successfully addressed all the major concerns I previously had. There are only a few minor language issues I noticed while re-reading, which I detail below:

p. 6: At the top of the page, the reference to the McHaney et al. paper is numbered as 10, but should be 12.

p. 8: "We will assess generalization in each session by presenting learners with novel stimuli spoken by novel talkers that they do not encounter during training and never receive feedback about the correct category." This seems ungrammatical at the end. Maybe "…that they do not encounter during training, without providing feedback about the correct category" would be better?

p. 11: "…without making sacrifices in performance" should perhaps be "without making sacrifices in accuracy" (since response time is also part of performance).

p. 11: The reference to the McHaney et al. paper is numbered wrongly again.

p. 15: Line 319 starts with "Click or tap here to enter text."

In addition, I thank the authors for attempting to explain the validity of their original model. I'm genuinely interested in how this would work, and though I don't really understand the argument made in the response letter, I'll look forward to seeing their future work on this topic. To that end, if I were a hypothetical reviewer of that work, one thing I'd definitely want to see would be model recovery simulations that show that the more flexible model is selectively preferred when data are generated by a model with boundaries that change within a trial (but not when they are generated by a model without such a feature).

Reviewer #3: The authors have addressed my comments and concerns in their revision. I have one small suggestion for a further analysis of drift rates that the authors might wish to consider (though I don’t think it is essential). It may provide some further insight into the extent to which WM relates to extraction of evidence in general vs. more focused extraction of categorically diagnostic information.

1 - I understand the focus on drift rates for the correct response option (i.e., when stimulus and response match). This indexes absolute evidence accumulation for the one option. It would be interesting to know whether there are discriminability differences as well, however. If correct drift rates were normalized across the sum across accumulators, you could further consider discriminability of the response options. This is potentially important because it pertains to whether higher WM is related to better extraction of evidence overall vs. extraction of diagnostic evidence that differentiates the categories.

Line 319 – Editing issue, “Click or tap here to enter text”.

Lines 319-320 – Was RT filtering based on slowest/fastest 1% of responses in the entire data set, or was filtering done separately for each participant?

7. PLOS authors have the option to publish the peer review history of their article (what does this mean?). If published, this will include your full peer review and any attached files.

Reviewer #1: No

Reviewer #3: No

---

## [Author Response · Author response to Decision Letter 2]

12 Jan 2024

We thank the editor and reviewers for consideration of our revised manuscript. We respond to each point raised by the reviewers with our responses in blue italics. 

Reviewer #1

The authors' current version of the manuscript has successfully addressed all the major concerns I previously had. There are only a few minor language issues I noticed while re-reading, which I detail below:

R1.C1: p. 6: At the top of the page, the reference to the McHaney et al. paper is numbered as 10, but should be 12.

We have fixed this.

R1.C2: p. 8: "We will assess generalization in each session by presenting learners with novel stimuli spoken by novel talkers that they do not encounter during training and never receive feedback about the correct category." This seems ungrammatical at the end. Maybe "…that they do not encounter during training, without providing feedback about the correct category" would be better?

We have made the suggested change. 

R1.C3: p. 11: "…without making sacrifices in performance" should perhaps be "without making sacrifices in accuracy" (since response time is also part of performance).

We have made the suggested change.

R1.C4: p. 11: The reference to the McHaney et al. paper is numbered wrongly again.

We have fixed this.

R1.C5: p. 15: Line 319 starts with "Click or tap here to enter text."

We have fixed this.

R1.C6: In addition, I thank the authors for attempting to explain the validity of their original model. I'm genuinely interested in how this would work, and though I don't really understand the argument made in the response letter, I'll look forward to seeing their future work on this topic. To that end, if I were a hypothetical reviewer of that work, one thing I'd definitely want to see would be model recovery simulations that show that the more flexible model is selectively preferred when data are generated by a model with boundaries that change within a trial (but not when they are generated by a model without such a feature).

We thank the reviewer for their comments – this will surely help us in preparing this future work for publication. 

Reviewer #3

The authors have addressed my comments and concerns in their revision. I have one small suggestion for a further analysis of drift rates that the authors might wish to consider (though I don’t think it is essential). It may provide some further insight into the extent to which WM relates to extraction of evidence in general vs. more focused extraction of categorically diagnostic information.

R3.C1: 1 - I understand the focus on drift rates for the correct response option (i.e., when stimulus and response match). This indexes absolute evidence accumulation for the one option. It would be interesting to know whether there are discriminability differences as well, however. If correct drift rates were normalized across the sum across accumulators, you could further consider discriminability of the response options. This is potentially important because it pertains to whether higher WM is related to better extraction of evidence overall vs. extraction of diagnostic evidence that differentiates the categories.

We appreciate this comment but see this analysis as outside of the scope of the current investigation. We have made a note about pursuing this possibility in future work. 

R3.C2: Line 319 – Editing issue, “Click or tap here to enter text”.

We have fixed this.

R3.C3: Lines 319-320 – Was RT filtering based on slowest/fastest 1% of responses in the entire data set, or was filtering done separately for each participant?

We have clarified this (line 323-324, page 15): The data were filtered to exclude very fast and very slow responses by removing the top and bottom 1% of all trials across all participants based on reaction time.

---

## [Editor Report · Decision Letter 3]

16 Jan 2024

Individual differences in working memory impact the trajectory of non-native speech category learning

PONE-D-23-00404R3

Dear Dr. Roark,

We’re pleased to inform you that your manuscript has been judged scientifically suitable for publication and will be formally accepted for publication once it meets all outstanding technical requirements.

Kind regards,

Alessandra S. Souza, Ph.D.

Academic Editor

PLOS ONE